# Shared and distinct mechanisms of UBA1 inactivation across different diseases

Jason C Collins [ID][1], Samuel J Magaziner[2,3], Maya English [ID][1], Bakar Hassan[4], Xiang Chen[4], Nicholas Balanda [ID][2,3], Meghan Anderson[2,3], Athena Lam[2,3], Sebastian Fernandez-Pol [ID][5], Bernice Kwong[6], Peter L Greenberg[7], Benjamin Terrier[8], Mary E Likhite[2,3], Olivier Kosmider[9], Yan Wang [ID][10], Nadine L Samara [ID][11], Kylie J Walters [ID][4], David B Beck[2,3,12,13] & Achim Werner [ID][1,13 ✉]

## Abstract

**Most cellular ubiquitin signaling is initiated by UBA1, which activates and transfers ubiquitin to tens of E2 enzymes. Clonally acquired *UBA1* missense mutations cause an inflammatory-hematologic overlap disease called VEXAS (vacuoles, E1, X-linked, autoinflammatory, somatic) syndrome. Despite extensive clinical investigation into this lethal disease, little is known about the underlying molecular mechanisms. Here, by dissecting VEXAS-causing *UBA1* mutations, we discovered that p.Met41 mutations alter cytoplasmic isoform expression, whereas other mutations reduce catalytic activity of nuclear and cytoplasmic isoforms by diverse mechanisms, including aberrant oxyester formation. Strikingly, non-p.Met41 mutations most prominently affect transthioesterification, revealing ubiquitin transfer to cytoplasmic E2 enzymes as a shared property of pathogenesis amongst different VEXAS syndrome genotypes. A similar E2 charging bottleneck exists in some lung cancer-associated *UBA1* mutations, but not in spinal muscular atrophy-causing *UBA1* mutations, which instead, render UBA1 thermolabile. Collectively, our results highlight the precision of conformational changes required for faithful ubiquitin transfer, define distinct and shared mechanisms of UBA1 inactivation in diverse diseases, and suggest that specific E1-E2 modules control different aspects of tissue differentiation and maintenance.**

**Keywords** UBA1; Ubiquitin; VEXAS Syndrome; Spinal Muscular Atrophy (SMA); Lung Cancer in Never Smokers (LCINS)
**Subject Categories** Molecular Biology of Disease; Post-translational Modifications & Proteolysis

## Introduction

Ubiquitylation is an essential posttranslational modification that controls virtually all cellular processes by regulating protein degradation and signaling pathways (Dikic and Schulman, 2023; Oh et al, 2018; Werner et al, 2017). The majority of ubiquitylation is initiated by the E1 enzyme UBA1, which is encoded on the X-chromosome and is produced as nuclear UBA1a (from p.M1) and cytoplasmic UBA1b (from p.M41) isoforms by alternative translation initiation (Handley-Gearhart et al, 1994a). UBA1 function depends on its ability to activate and transfer ubiquitin to tens of E2 enzymes that further collaborate with hundreds of E3 ligases to modify and determine the fate of thousands of cellular substrates (Beck et al, 2022; Schulman and Harper, 2009; Stewart et al, 2016). While regulation of ubiquitin signaling at the level of E2/E3 interplay or E3/substrate recruitment is well understood, far less is known about whether control of ubiquitin activation and transfer to E2 enzymes contributes to physiological processes. Such regulation has been suggested by recent reports showing that mutations in UBA1 can lead to distinct human diseases (Beck et al, 2020; Ramser et al, 2008; Zhang et al, 2021), but the underlying molecular principles remain unclear.

At a structural level, UBA1 is a multidomain protein that undergoes large conformational changes to dynamically remodel catalytic centers for ubiquitin adenylation, thioester formation, and transthiolation reactions (Hann et al, 2019; Lee and Schindelin, 2008; Lorenz et al, 2013; Lv et al, 2017a; Lv et al, 2018; Lv et al, 2017b; Schafer et al, 2014; Williams et al, 2019; Yuan et al, 2021). Initially, a ubiquitin and ATP are bound to the active adenylation domain (AAD), and a ubiquitin adenylate is formed. This first adenylation reaction occurs with the second catalytic cysteine half (SCCH) domain in an open conformation, which then closes to bring UBA1's catalytic cysteine residue proximal to the C-terminus of ubiquitin for thioester formation (Hann et al, 2019).

[1]Stem Cell Biochemistry Section, National Institute of Dental and Craniofacial Research, National institutes of Health, Bethesda, MD, USA. [2]Center for Human Genetics and Genomics, New York University School of Medicine, New York, NY, USA. [3]Division of Rheumatology, Department of Medicine, New York University School of Medicine, New York, NY, USA. [4]Protein Processing Section, Center for Structural Biology, National Cancer Institute, National Institutes of Health, Frederick, MD, USA. [5]Department of Pathology, Stanford University Medical School, Stanford, CA, USA. [6]Department of Dermatology, Stanford University Cancer Center, Stanford, CA, USA. [7]Division of Hematology, Stanford University Cancer Center, Stanford, CA, USA. [8]Department of Internal Medicine, Hôpital Cochin, Assistance Publique-Hôpitaux de Paris, Paris, France. [9]Laboratory of Hematology, Hôpital Cochin, Assistance Publique-Hôpitaux de Paris, Paris, France. [10]Mass Spectrometry Facility, National Institute of Dental and Craniofacial Research, National Institutes of Health, Bethesda, MD, USA. [11]Structural Biochemistry Unit, National Institute of Dental and Craniofacial Research, National Institutes of Health, Bethesda, MD, USA. [12]Department of Biochemistry and Molecular Pharmacology, New York University School of Medicine, New York, NY, USA. [13]These authors contributed equally: David B Beck, Achim Werner. ✉E-mail: achim.werner@nih.gov

Subsequently, the SCCH domain opens, allowing a second ubiquitin molecule to bind, giving rise to a doubly loaded UBA1 complex with one ubiquitin bound non-covalently to the adenylation domain ($Ub^A$) and another ubiquitin thioester-linked to the catalytic cysteine residue ($Ub^T$) contacting the first catalytic cysteine half (FCCH) domain. Compared to singly loaded UBA1, this doubly loaded UBA1 complex is more active in transferring ubiquitin to E2 enzymes (Haas et al, 1988; Pickart et al, 1994). During transthiolation, an E2 enzyme binds to the ubiquitin fold domain (UFD) of UBA1 that transitions from an open to a closed confirmation to bring the E1 and E2 active sites in close proximity (Lv et al, 2017b; Olsen and Lima, 2013; Williams et al, 2019; Yuan et al, 2021). Thus, a series of highly coordinated intramolecular movements within UBA1 are required to ensure faithful ubiquitin activation and transfer. Yet, whether and how defects in these conformational changes can lead to human disease is not well understood.

Somatic mutations in *UBA1* cause a late-onset autoinflammatory disease, which we have termed VEXAS (Vacuole, E1 enzyme, X-linked, Autoinflammatory, Somatic) syndrome (Beck et al, 2020). Pathogenic *UBA1* mutations result in an often-fatal hematologic and inflammatory symptoms, which can present with overlapping features of well-established clinical diagnoses such as myelodysplastic syndrome (MDS) and vasculitis (Ferrada et al, 2021; Georgin-Lavialle et al, 2022). VEXAS syndrome and *UBA1* pathogenic mutations are present in 1/4000 males over the age of 50 and represent a significant public health burden (Beck et al, 2023). Effective treatments are limited by our lack of understanding of how *UBA1* mutations result in disease.

The predominant VEXAS syndrome-causing mutations identified to date occur at p.M41, the start codon for the cytoplasmic isoform of UBA1 (UBA1b). These mutations lead to an isoform swap with loss of active UBA1b and emergence of a new catalytically impaired isoform, UBA1c, initiated from p.M67 (Beck et al, 2020). However, the relative contributions of loss of UBA1b versus gain of UBA1c in terms of disease pathogenesis have remained unclear. In addition, recent reports, including ours, suggest that mutations at other sites than p.Met41 can cause VEXAS syndrome, highlighting the existence of disease mechanisms distinct from the canonical loss of cytoplasmic UBA1 activity, but molecular details have remained elusive (Faurel et al, 2023; Poulter et al, 2021; Sakuma et al, 2023; Stiburkova et al, 2023).

Beyond VEXAS syndrome, distinct *UBA1* mutations have been implicated in lung cancer in never smokers (LCINS) and spinal muscular atrophy (SMA) (Ramser et al, 2008; Zhang et al, 2021). In LCINS, *UBA1* heterozygous frameshift, non-sense, and missense mutations were identified as potential somatic drivers of malignancy in females. In SMA, rare germline, hemizygous, missense mutations were identified in male children. Both genotypes are unique from VEXAS, either because variants are found exclusively in females in lung cancer, or germline in SMA, and no shared variants have been identified across disease types. It remains unclear whether any molecular mechanisms may be shared across *UBA1*-related disease and how specific phenotypes can be caused by alterations of a pleiotropic and ubiquitously expressed gene.

Here, we utilized unbiased screening of exome sequencing data from patients with similar inflammatory disorders to VEXAS without a known pathogenic mutation to identify novel disease-causing variants in *UBA1*. We systematically characterized them along with previously reported pathogenic VEXAS mutations both in vitro and in cells. We found that only p.Met41 mutations lead to a loss of UBA1b levels and production of UBA1c, while all other tested variants did not alter isoform expression or subcellular localization. Instead, these non-canonical VEXAS mutations reduce catalytic function of both cytoplasmic and nuclear isoforms by diverse mechanisms, including intramolecular auto-ubiquitylation via oxyesters, that most prominently affect the E2 transthiolation step. Strikingly, a similar bottleneck in E2 charging is present in some LCINS but not SMA mutations, which instead render UBA1 activity thermolabile.

Taken together, our results expand the genetic causes of VEXAS syndrome, demonstrate that UBA1c expression is not required for disease pathogenesis, and suggest loss of ubiquitin conjugation to cytoplasmic E2 enzymes as a shared property of pathogenesis amongst different VEXAS syndrome genotypes. In addition, our in vitro profiling approach reveals two major classes of UBA1 mutations that (i) bottleneck at the E2 transfer step and are deficient in E2 charging and polyubiquitylation in cells and (ii) render UBA1 activity thermolabile and exhibit subtle cellular defects. While LCINS comprise both classes, SMA mutations are exclusively thermolabile and non-canonical VEXAS mutations bottleneck at the E2 charging step, suggesting distinct and shared mechanisms of UBA1 inactivation across different disease states. These molecular insights provide a framework for studying regulatory principles of ubiquitylation at the E1-E2 level, for determining cellular downstream consequences of loss of UBA1 activity, and for developing therapeutic strategies to treat VEXAS syndrome and *UBA1*-related diseases in the future.

# Results

## Identification of novel somatic UBA1 mutations that cause VEXAS syndrome by a non-canonical mechanism

To better understand the mutational spectrum of VEXAS syndrome, we analyzed *UBA1* variants within a cohort of patients referred for genetic testing given clinical concern for severe systemic inflammatory disease. 183 total samples were received over a 6-month period from older male individuals, primarily of peripheral blood. Through Sanger sequencing, we identified 76 individuals carrying known pathogenic missense p.Met41Val/Thr/Leu and splice-site altering mutations in *UBA1* (Fig. 1A). In addition, we found one patient with a p.Tyr55His mutation, at a site adjacent to a known pathogenic variant (p.Ser56Phe) (Fig. EV1A). The remaining 106 undiagnosed patients all underwent exome sequencing. Through this approach, we uncovered 1 *UBA1* p.Met41Val mutation with a low variant allele fraction (VAF) of 9% that was confirmed by digital droplet PCR (ddPCR). We next cross compared *UBA1* variants in our cohort of 106 individuals with 58 individuals from an international center (Cochin Hospital, Paris, France) referred for similar inflammatory manifestations to VEXAS who tested negative for canonical VEXAS mutations by Sanger-sequencing, filtering for shared, rare, and predicted deleterious mutations. We identified two recurrently mutated amino acids in UBA1, both highly conserved through evolution, and confirmed the presence of these somatic p.Ala478Ser and p.Asp506Asn/Gly variants by ddPCR (Fig. EV1B,C). Such

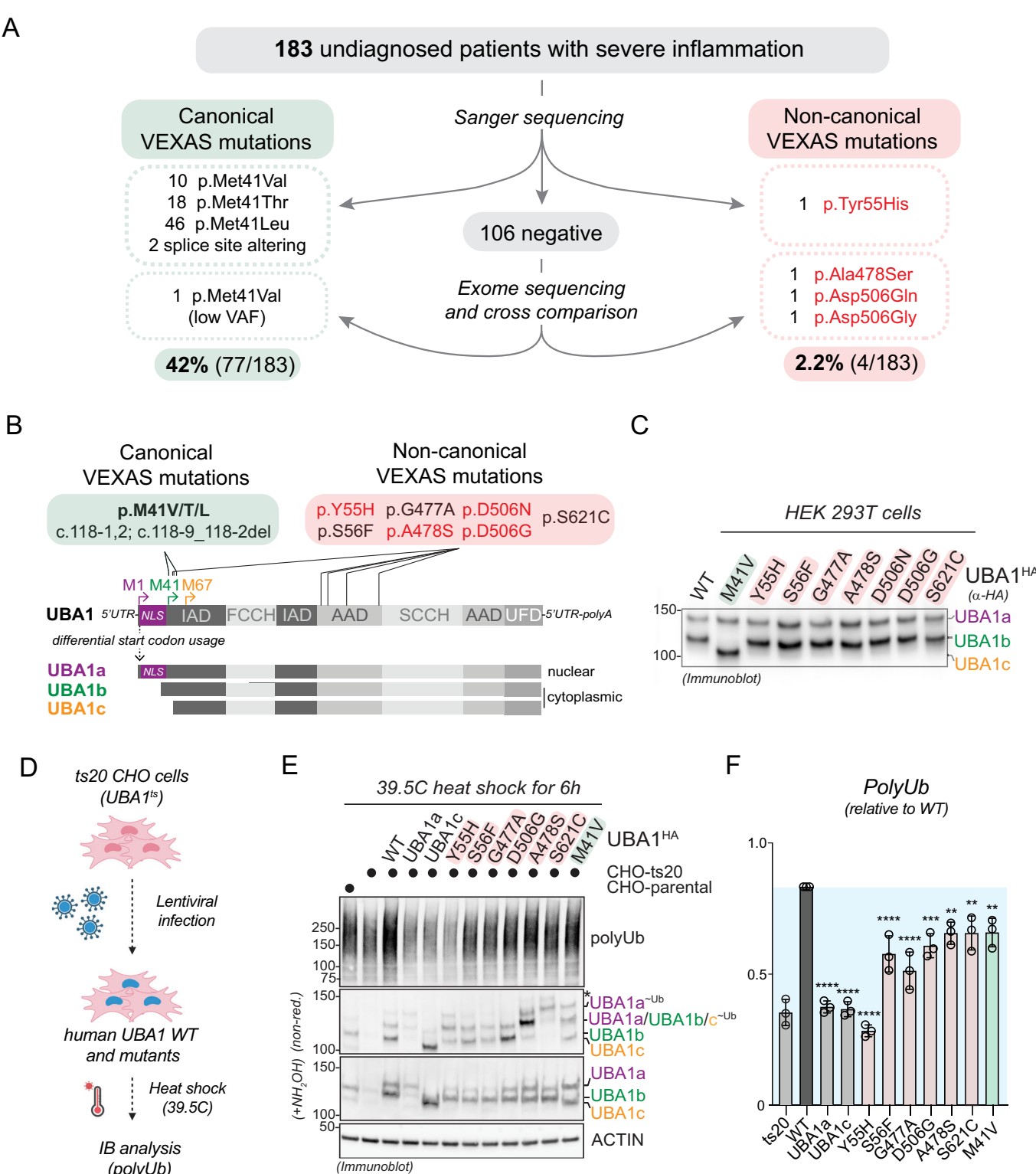

analysis provided three additional patient diagnoses in the NIH cohort. Taken together, our single-center study for VEXAS testing revealed 42% (77/183) of referred patients to carry known pathogenic *UBA1* mutations and for 2.2% (4/183) of the cohort we found potential disease-causing *UBA1* mutations by either Sanger or exome sequencing. Patients with these novel somatic *UBA1* mutations had highly similar overlapping disease to canonical VEXAS syndrome, including inflammatory and hematologic diagnoses, anemia, elevated inflammatory markers, and steroid dependence (Tables 1 and EV1; Fig. EV1D,E). We chose to study these four novel mutations (p.Tyr55His, p.Ala478Ser, p.Asp506Asn and p.Asp506Gly) in detail, along with three other

**Figure 1. Identification of novel somatic UBA1 mutations that cause VEXAS syndrome by a non-canonical mechanism.**

(A) Systematic analyses of *UBA1* in a cohort of undiagnosed inflammatory patients identified canonical VEXAS mutations in 42% (77/183) of patients and non-canonical VEXAS mutations (not affecting p.Met41) in 2.2% (4/183) of patients. (B) Schematic overview of UBA1 mRNA and protein isoforms that are produced through alternative start codon usage. Canonical VEXAS mutations (green box) reduce the translation of the cytoplasmic isoform UBA1b (initiated from p.M41, highlighted in green) and result in expression of a shorter, catalytically impaired isoform UBA1c (initiated from p.M67, highlighted in orange). Non-canonical VEXAS mutations identified previously (in red box) or in this study (highlighted in red) do not affect p.M41 and are positioned in the inactive adenylation domain (IAD) or active adenylation domain (AAD). (C) Non-canonical VEXAS mutations do not lead to the cytoplasmic isoform swap from UBA1b to UBA1c, as evidenced by immunoblotting of HEK293T cell lysates transfected with the indicated UBA1 variants. (D) Workflow of a cellular assays to test for a functional impact of UBA1 variants in cells. Chinese hamster ovary (CHO) cells carrying a temperature sensitive UBA1 allele are lentivirally transfected with indicated UBA1 variants, incubated at the restrictive temperature, and lysates are analyzed by immunoblotting. (E) Immunoblots of CHO cells treated according to the flow diagram in panel (D), revealing that similar to the canonical VEXAS mutation p.M41V, non-canonical VEXAS mutations are impaired in their ability to support ubiquitylation in CHO cells, thus suggesting that they compromise the catalytic activity of UBA1. * = UBA1 ubiquitin oxyesters as demonstrated in Fig. 3. (F) Quantification of global polyubiquitylation levels shown in panel (E). $n = 3$ biological replicates, mean $-/+$ s.d., **$p < 0.01$, ***$p < 0.001$, ****$p < 0.0001$, one-way ANOVA. Source data are available online for this figure.

**Table 1. Clinical table of patients with non-canonical VEXAS mutations.**

| Patient ID | P1 | P2 | P3 | P4 | P5 | P6 |
|---|---|---|---|---|---|---|
| UBA1 Mutation | p.Ala478Ser | p.Ala478Ser | p.Asp506Asn | p.Asp506Gly | p.Met41Val; p.Ala478Ser | p.Tyr55His |
| Sex (M/F) | M | M | M | M | M | M |
| Deceased (Y/N) | N | N | N | Y | N | N |
| *Labs:* | | | | | | |
| Elevated inflammatory markers | Y | Y | Y | Y | Y | Y |
| Anemia (Y/N) | Y | Y | Y | Y | Y | Y |
| Fevers (Y/N) | N | Y | Y | Y | Y | Y |
| Skin Involvement (Y/N) | Y | Y | N | Y | Y | Y |
| Pulmonary infiltrate (Y/N) | N | N | N | Y | Y | N |
| Vasculitis (Y/N) | N | N | Y | Y | N | Y |
| Ear/Nose chondritis (Y/N) | N | N | N | N | Y | N |
| Bone marrow diagnosis | MDS | MDS | NA | MDS | None | MDS |
| Treatment History (drug and effect): | | | | | | |
| Glucocorticoids (Y/N) | Y, GC dependency | Y, GC dependency | Y, GC dependency | Y, GC dependency | Y, GC dependency | Y, GC dependency |

recently reported variants in the literature, namely p.Ser56Phe, p.Gly477Ala, and p.Ser621Cys (Beck et al, 2023; Faurel et al, 2023; Poulter et al, 2021; Stiburkova et al, 2023) with the goal of better understanding how UBA1 function is altered in VEXAS syndrome (Fig. 1B). An overview of all UBA1 mutations investigated in this study can be found in the supplement (Table EV2).

We first tested the impact of non-canonical VEXAS mutations on UBA1 isoform translation upon ectopic expression in cells. As expected from their locations being distinct from the start codon of cytoplasmic UBA1b and opposed to the p.M41V variant, non-canonical VEXAS mutations had no obvious impact on UBA1 isoform production and did not result in the cytoplasmic isoform swap from UBA1b to UBA1c (Fig. 1B,C). In addition, non-canonical VEXAS mutation did not markedly affect UBA1 subcellular localization (Appendix Fig. S1). To verify that these novel mutations would indeed have an impact on UBA1 activity, we established a cellular rescue system based on the ts20 CHO cell line carrying a temperature sensitive UBA1 allele (Handley-Gearhart et al, 1994b; Kulka et al, 1988) (Fig. 1D). Upon incubation of these cells at the restrictive temperature for 6 h and as compared to their parental counterparts, endogenous UBA1 is degraded, resulting in reduction of total cellular polyubiquitylated proteins

(Fig. 1E, lane 1 versus 2). We found that loss of polyubiquitylation can be rescued in these cells by lentiviral reconstitution with human wild-type (WT) UBA1, but not with constructs only encoding for only UBA1a or UBA1c (Fig. 1E, lane 2 versus 3, 4 and 5). Importantly, similar to the canonical VEXAS mutation p.M41V, none of the non-canonical VEXAS mutations were able to support polyubiquitylation to the same extent as wild-type UBA1 (Fig. 1E,F). Taken together, these results demonstrate that non-canonical VEXAS mutations do not specifically reduce cytoplasmic ubiquitylation by an isoform swap, and suggest that they reduce the catalytic activity of both nuclear and cytoplasmic pools of UBA1.

## Non-canonical VEXAS mutations are clustered around the ATP-binding site of UBA1 and a subset exhibits small reductions in ubiquitin adenylation and thioester formation

Analysis of previously reported crystal structures of human and yeast UBA1 revealed that non-canonical VEXAS mutations are clustered close to the binding site for ubiquitin and ATP in the adenylation site (Fig. 2A) (Lv et al, 2018; Olsen and Lima, 2013). In particular, A478, D506, and S56 participate in positioning ATP in

    

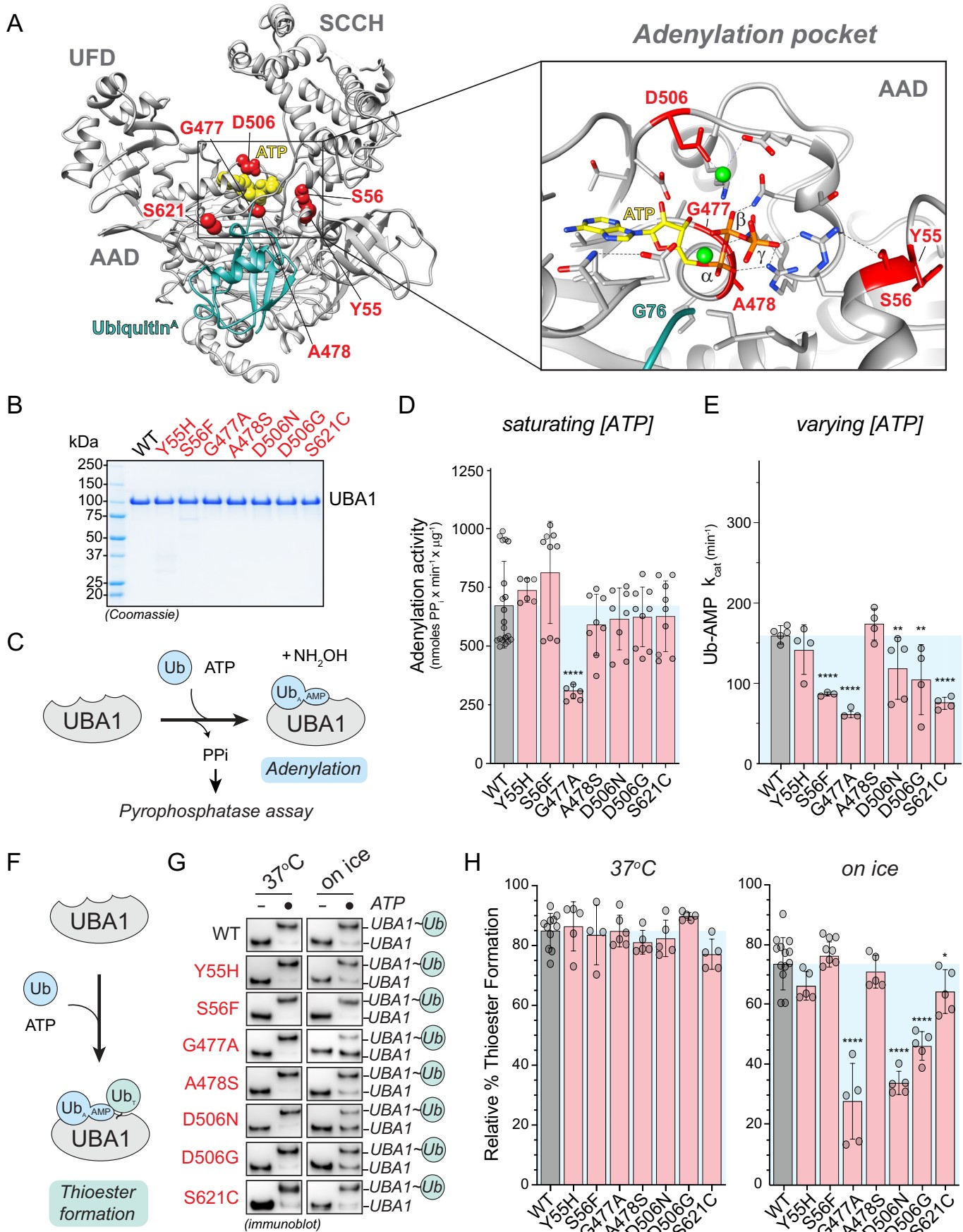

Figure 2. Non-canonical VEXAS mutations cluster around UBA1's ATP-binding site and a subset causes small reductions in ubiquitin adenylation and thiolation.

(A) Structure of human UBA1 (gray) with ubiquitin (cyan) at the adenylation site (PDB: 6DC6) highlighting where non-canonical VEXAS mutations occur (red). An expansion of the adenylation pocket is included (right). ATP (yellow spheres (left) or stick view (right)) was modeled into the structure using an S. pombe UBA1-Ub/ ATP•Mg structure (PDB:4II3). Residues substituted by non-canonical VEXAS mutations are highlighted in red (spheres (left) or stick view (right)), revealing that they cluster around the adenylation site and participate in orienting ATP and $Mg^{+2}$ (green spheres) in the catalytic pocket. To clearly visualize the ATP binding pocket in the expanded view, the cross over loop containing the S621 is not shown. All stick representations: carbon (gray), oxygen (red), nitrogen (blue), and phosphorus (orange). (B) Coomassie gel of WT UBA1b$^{His}$ and indicated non-canonical VEXAS mutants (red) recombinantly expressed and purified from E. coli, revealing high purity. (C) Schematic overview of the in vitro pyrophosphatase assay used to determine the efficiency of the adenylation reaction of UBA1. (D) Only UBA1 p.G477A, but no other non-canonical VEXAS variant significantly affects adenylation activity in vitro at saturating ATP and ubiquitin conditions. $n = 2–3$ biological replicates with 3 technical replicates each, mean $−/+$ s.d., $****p < 0.0001$, one-way ANOVA. (E) Michaelis-Menten kinetics for ATP usage showed -2-fold deficiency in forming the Ub-AMP in UBA1 p.S56F, p.G477A, and p.S621C. The rate of Ub-AMP formation ($k_{cat}$) was calculated for $n = 3–5$ biological as indicated, mean $−/+$ s.d., $**p < 0.01$, $****p < 0.0001$, one-way ANOVA. (F) Schematic overview of the UBA1 thioester formation assay. (G) Non-reducing immunoblot analysis of in vitro ubiquitin thioester formation reactions carried out at 37 °C (left panel) or on ice (right panel). Reactions on ice reveal defects for UBA1 p.G477A, p.D506N, p.D506G. (H) Quantification of thioester formation percentage (UBA1-Ub/total UBA1 signal) of the experiments shown in panel (G). $n = 5–10$ biological replicates, mean $−/+$ s.d., $*p < 0.05$, $****p < 0.0001$, one-way ANOVA. Source data are available online for this figure.

the catalytic pocket and G477 is within 4 Å of the α-phosphate. We thus reasoned that substitutions at these residues could interfere with ATP binding and adenylation. To test this hypothesis, we purified recombinant WT and mutant UBA1b (aa 41–1058) from E. coli (Fig. 2B) and measured their ability to catalyze the first adenylation reaction using a previously described pyrophosphatase assay (Fig. 2C) (Balak et al, 2017). Surprisingly, at saturating substrate concentrations, only the p.G477A mutant exhibited a ~2-fold reduction in ubiquitin adenylation activity, while all other non-canonical VEXAS mutants behaved similarly to the WT protein (Fig. 2D). To test for more subtle defects of these mutants that might occur under limiting substrate concentrations, we next varied the ATP concentrations and determined the rate of Ub-AMP formation ($k_{cat}$). Under these conditions, we found that p.S56F, p.G477A, and p.S621C were up to ~2-fold deficient in forming the Ub-AMP, while the remaining variants showed no obvious difference to WT UBA1 (Fig. 2E). We thus tested whether the non-canonical VEXAS mutations more strongly affect ubiquitin thioester formation and incubated recombinant UBA1 proteins with saturating concentrations of ATP and ubiquitin (Fig. 2F). While all non-canonical VEXAS mutations behaved similarly to WT UBA1 at 37 °C, we again detected ~2-fold reductions in ubiquitin thiolation for a subset of mutations (p.G477A, p.D506N/ G) when the reactions were slowed down by performing them on ice (Fig. 2G,H). Taken together, these results indicate that select non-canonical VEXAS mutations have reductions in ubiquitin adenylation and/or thiolation. Yet, these defects are only in the range of up to ~2-fold and some variants (i.e., p.Y55H, p.A478S) showed no differences compared to WT UBA1, suggesting other mechanisms of UBA1 inactivation.

## UBA1 p.S621C and p.A478S form oxyesters by distinct mechanisms

In rescue experiments with CHO cells reconstituted with different UBA1 protein variants, we observed unexpected higher molecular weight bands for the UBA1 p.A478S and p.S621C variants when the lysates were subjected to immunoblot analysis under non-reducing conditions (Fig. 1E). These species were also present when ectopically expressing UBA1 mutants in HEK293T cells and they are consistent in size with a second ubiquitin conjugated to UBA1a and UBA1b, respectively (Fig. 3A). We could not reduce these conjugates with β-mercaptoethanol (β-ME) but could with

hydroxylamine ($NH_2OH$), indicative of oxyesters. The aberrant species identified were dependent on thioester formation by the catalytic cysteine residue, as C632A mutants did not support their formation (Fig. 3B). In addition, oxyesters for UBA1 p.A478S and p.S621C differed in their gel migration pattern (Fig. 3A,B) and we only observed the formation of additional species for UBA1 p.S621C but not p.A478S at the chosen conditions of our in vitro assays (Fig. 3C). Together, these results suggest that during thiolation, UBA1 p.A478S and p.S621C form aberrant oxyesters by distinct mechanisms.

We first sought to understand the mechanism of oxyester formation of UBA1 p.A478S. Modeling the A478S mutation in an available UBA1 structure bound to Ub-AMP (Schafer et al, 2014) revealed that the hydroxyl-group is in optimal position to attack the phospho-anhydride bond of the ubiquitin adenylate (Fig. 3D), strongly suggesting that the serine at this position forms the oxyester. Indeed, we could detect the oxyester at p.A478S by mass spectrometry analysis of immunoprecipitated UBA1 from HEK293T cells (Figs. 3E and EV2A,B) and introduction of cysteine at the position does not support double loading or formation of the higher molecular weight species (Fig. 3F). Given these results in cells and the fact that the hydroxyl-group of p.A478S is in optimal position to form the oxyester (Fig. 3D), we next wondered why we did not observe doubly conjugated UBA1 species in vitro (Fig. 3C). We hypothesized that additional cellular factors may promote oxyester formation and thus tested for the role of E2 enzymes. Intriguingly, addition of the E2 enzyme UBE2D3 to UBA1 pre-charged with ubiquitin led to the formation of an oxyester with UBA1 p.A478S but not the wild-type protein in vitro (Fig. 3G). Molecular dynamics simulations of UBA1 bound to ubiquitin and ATP in the absence and presence of an E2 enzyme revealed that the E2 enzyme reduces intramolecular movements in the UBA1 AAD, which binds the ubiquitin and ATP (Fig. EV2C). Together these findings suggest that oxyester formation of UBA1 p.A478S occurs when a doubly loaded UBA1 binds an E2 enzyme, which likely stabilizes the intermolecular positioning of the Ub-AMP in the binding site, thus promoting the attack of the ubiquitin adenylate phospho-anhydride bond by the hydroxyl-group of A478S (Fig. 3H).

While UBA1 pA478S forms an oxyester directly at the mutation site, the location of the oxyester in UBA1 p.S621C was less clear, as the mutation results in loss of a hydroxyl-group and gain of a sulfhydryl-group. Interestingly, we noted that S621 is in the cross-

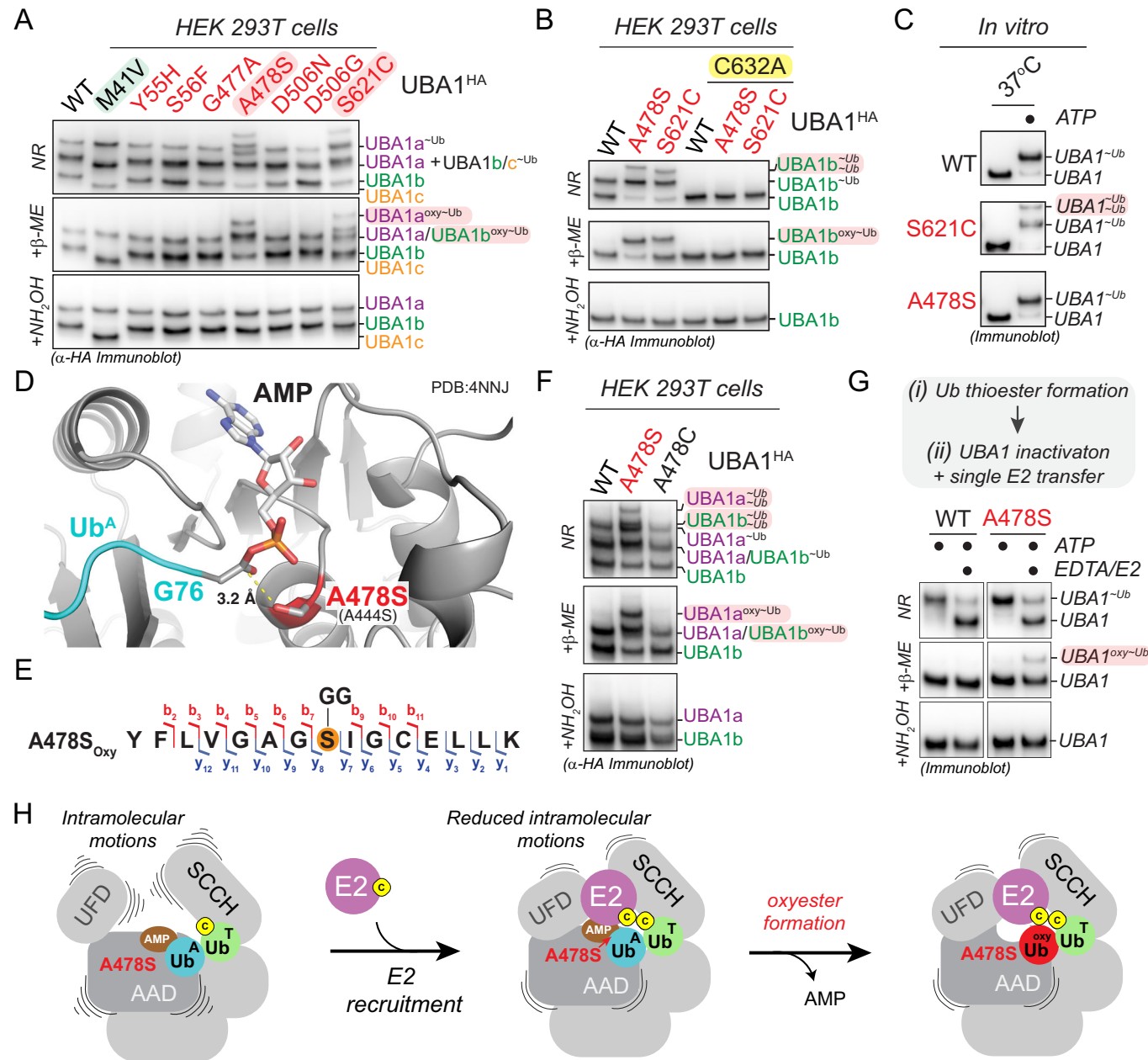

over loop, a structural element that is known to be dynamically remodeled during UBA1 thiolation (Lv et al, 2017b) and that contains several serine residues near S621 (Fig. 4A). This suggests that the oxyester could be forming on a neighboring serine residue utilizing the substituted sulfhydryl-group of S621C as an intermediate. Indeed, mass spectrometry analysis of immunoprecipitated UBA1 from HEK293T cells localized the oxyester to S619 (Figs. 4B and EV3A,B). We could verify this observation by mutational analyses, as alanine substitution of S619 but none of the other neighboring serine residues eliminated the oxyester bond and rendered the additional UBA1 species sensitive to β-ME (Fig. 4C), suggesting buildup of the thioester intermediate at S621C. Together with our finding that formation of aberrant ubiquitin species depends on an intact catalytic cysteine residue (Fig. 3B), these

results demonstrate that oxyester formation at S619 is preceded by ubiquitin transthiolation of the catalytic cysteine residue to S621C.

As the thioester intermediates during oxyester formation at S619 are transient, we were not able to monitor their buildup and loss during in vitro reactions. However, to further understand why only S619 and none of the other nearby serine residues can attack the ubiquitin thioester at S621C, we performed molecular dynamics simulations investigating interactions with neighboring residues that could serve as catalytic bases during oxyester formation. We observed that amongst the serine residues in the cross-over loop, the intramolecular distances of only S619 and S620 with their respective neighbors D585 and E616 were indicative of stable hydrogen bonding (Fig. 4D,E). For both interactions, within the 100 ns molecular dynamics simulation, the average distance

**Figure 3.   UBA1 p.S621C and p.A478S form aberrant oxyesters by distinct mechanisms.**

(A) UBA1 p.S621C and p.A478S form aberrant ubiquitin conjugates in cells that are reducible by NH₂OH but not β-ME, indicative of oxyesters. Indicated UBA1ᴴᴬ variants (canonical VEXAS mutation M41 in green, non-canonical VEXAS mutations in red) were expressed in HEK293 T cells and lysates were either not treated (non-reducing, NR) or treated with indicated reducing agents and subjected to anti-HA immunoblotting. Aberrant UBA1 species are highlighted by a red box. (B) UBA1 p.S621C- and p.A478S-dependent oxyester formation requires the catalytic cysteine residue C632 (yellow). Indicated UBA1bᴴᴬ variants were expressed in HEK293T cells and lysates were either not treated (non-reducing, NR) or treated with indicated reducing agents and subjected to anti-HA immunoblotting. Additional UBA1 species are highlighted by a red box. (C) UBA1 p.S621C, but not UBA1 p.A478S, forms aberrant ubiquitin conjugates in vitro. Indicated recombinant UBA1b proteins (500 nM) were incubated with 10 μM ubiquitin and 5 mM ATP for 10 min at 37 °C followed by anti-UBA1 immunoblot analysis. Additional UBA1 species are highlighted by a red box. (D) Crystal structure of *S. cerevisiae* UBA1 (gray) bound to Ub-AMP (Ubᴬ, PDB: 4NNJ) was modeled with the p.A478S mutation (*S. cerevisiae* p.A444S), showing that the introduced hydroxyl-group is in optimal position to attack the phospho-anhydride bond of the ubiquitin adenylate (cyan). Human numbering is displayed in bold with *S. cerevisiae* in parentheses. All stick representations: carbon (gray), oxygen (red), nitrogen (blue), and phosphorus (orange). (E) UBA1 p.A478S forms an oxyester at the mutation site. C-terminally FLAG tagged UBA1 p.A478S was expressed in HEK293T cells, followed by anti-FLAG immunoprecipitation and mass spectrometry analysis. The ubiquitylated residue is colored in orange and labeled with a diGly remnant (GG). Detected b and y ions for the oxyester-containing peptide are highlighted in red and dark blue, respectively. (F) UBA1 p.A478C does not support formation of aberrant ubiquitin conjugates in cells. Indicated UBA1ᴴᴬ variants were expressed in HEK293T cells and lysates were either not treated (non-reducing, NR) or treated with indicated reducing agents and subjected to anti-HA immunoblotting. (G) UBA1 p.A478S forms oxyesters in vitro in the presence of an E2 enzyme. (i) 500 nM UBA1b WT or p.A478S were incubated with 10 μM ubiquitin and 5 mM ATP, followed by (ii) addition of 100 mM EDTA and 2 μM E2 enzyme (UBE2D3). Reactions were either not treated (non-reducing, NR) or treated with indicated reducing agents and subjected to anti-UBA1 immunoblotting. (H) Model of how UBA1 p.A478S forms oxyesters. Recruitment of an E2 enzyme to a doubly loaded UBA1 complex reduces intramolecular movements in the domains of UBA1 that bind the ubiquitin in the adenylation site (Ubᴬ), thereby promoting the attack of the phospho-anhydride bond of the ubiquitin adenylate by the hydroxyl-group of A478S. Catalytic cysteines of UBA1 and the E2 are highlighted in yellow. Source data are available online for this figure.

---

between the serine and the catalytic base was ~3 Å (Fig. 4F), which would enable proton transfer. To determine which pair would likely have the stronger proton transfer interaction under physiological conditions, the pKₐ values were calculated. This analysis revealed predicted pKₐ values for S619 and D585 of $7.68 \pm 0.27$ and $6.38 \pm 0.25$, respectively (Fig. 4G). These values predict that at physiological pH, D585 acts as a catalytic base and suggest a reaction mechanism that involves deprotonation of S619 by putative catalytic base D585 forming an oxyanion. The S619 oxyanion could subsequently engage in a nucleophilic attack of the thioester at C621 to form an oxyester. This predicted mechanism provides a molecular explanation for our experimental findings (Figs. 4C and EV3C). Interestingly, recent work has implicated D585 in VEXAS disease pathogenesis (Faurel et al, 2023), suggesting that this residue could also play a role during normal catalysis. We conclude that oxyester formation for the UBA1 p.S621C variant occurs in three steps: (i) formation of the thioester at the catalytic cysteine residue C632, (ii) intramolecular transthiolation to C621, and (iii) oxyester formation at S619 (Fig. 4H).

## Non-canonical VEXAS mutations exhibit a bottleneck in ubiquitin transfer to E2 enzymes and are deficient in supporting E2 ubiquitin thioester levels in cells

We next wished to understand how oxyester formation of p.A478S and p.S621C mutants would affect UBA1 activity. We reasoned that the most likely impact would be at the level of ubiquitin transfer to E2 enzymes. Thus, we devised a sequential, three-phase in vitro assay, in which we allowed for (i) complete charging of UBA1 with ubiquitin (ii) ubiquitin transthiolation to a model E2 enzyme (UBE2D3) in a single turnover reaction, and (iii) UBA1 thiolation and E2 transthiolation in multi-turnover reactions (Fig. 5A). For WT UBA1, we found that in the single turnover transthiolation reaction (phase ii) most of the ubiquitin is transferred from UBA1 to the E2 enzyme, which was supplied at 4-fold excess (Fig. 5B). At the end of the multi-turnover reaction (phase iii), both, UBA1 and the E2 enzyme, were almost completely charged. In contrast, both oxyester-forming UBA1 variants (p.A478S and p.S621C), were

impaired in their ability to transfer ubiquitin to the E2 enzyme, as evidenced by the reduced amounts of charged E2 enzyme in single and multi-turnover reactions (phase ii,iii). In addition, quantification of multi-turnover reactions (phase iii) revealed a significant ~2–4-fold reduction in ubiquitin transfer to the E2 enzyme (Fig. 5C), while UBA1 charging was unaffected (Fig. 5D), suggesting that loss of ubiquitin transfer to the E2 enzyme is the major molecular defect in UBA1 p.A478S and p.S621C variants (Fig. 5E). Intriguingly, multi-transfer reactions revealed that for all non-canonical VEXAS mutations, the relative defects during transthiolation exceeded those of the thioester formation step (Fig. 5C–E). In addition, we observed defects of similar magnitude for all non-canonical VEXAS mutations when performing in vitro transthiolation assays with FITC-labeled ubiquitin and E2 enzymes that only contain a UBC domain (UBE2D3), have an additional N-terminal extension (UBE2E1), or an additional C-terminal extension (UBE2R2, UBE2S) (Figs. 5F and EV4A–C). Taken together, these in vitro findings suggest that non-canonical VEXAS mutations inactivate UBA1 by creating a bottleneck at the E2 transfer step (Fig. 5G).

We next analyzed the ubiquitin charging state of select E2s in CHO cells reconstituted with non-canonical VEXAS mutations. Consistent with our in vitro results, we observed that most non-canonical VEXAS mutations showed reduced ubiquitin thioester levels of select E2 enzymes (Figs. 5H and EV4D–G). However, in contrast to our in vitro results, charging of the tested E2s was differentially affected by specific non-canonical VEXAS variants. In addition, CHO cells expressing the UBA1 p.S621C mutant exhibited no obvious defects in the levels of ubiquitin thioester charged UBE2D3, UBE2C, and UBE2R2. To reconcile this finding with the observed reduction of polyubiquitylation levels in these cells (Fig. 1E,F), we tested additional E2 enzymes and identified UBE2S and UBE2L3 to be reduced in their charging state (Fig. 5I). While we cannot assess ubiquitin discharge from E2 enzyme in this system, these results strongly suggest that non-canonical VEXAS mutations exhibiting a bottleneck in E2 transfer in vitro (Fig. 5G) are also deficient in E2 charging in cells.

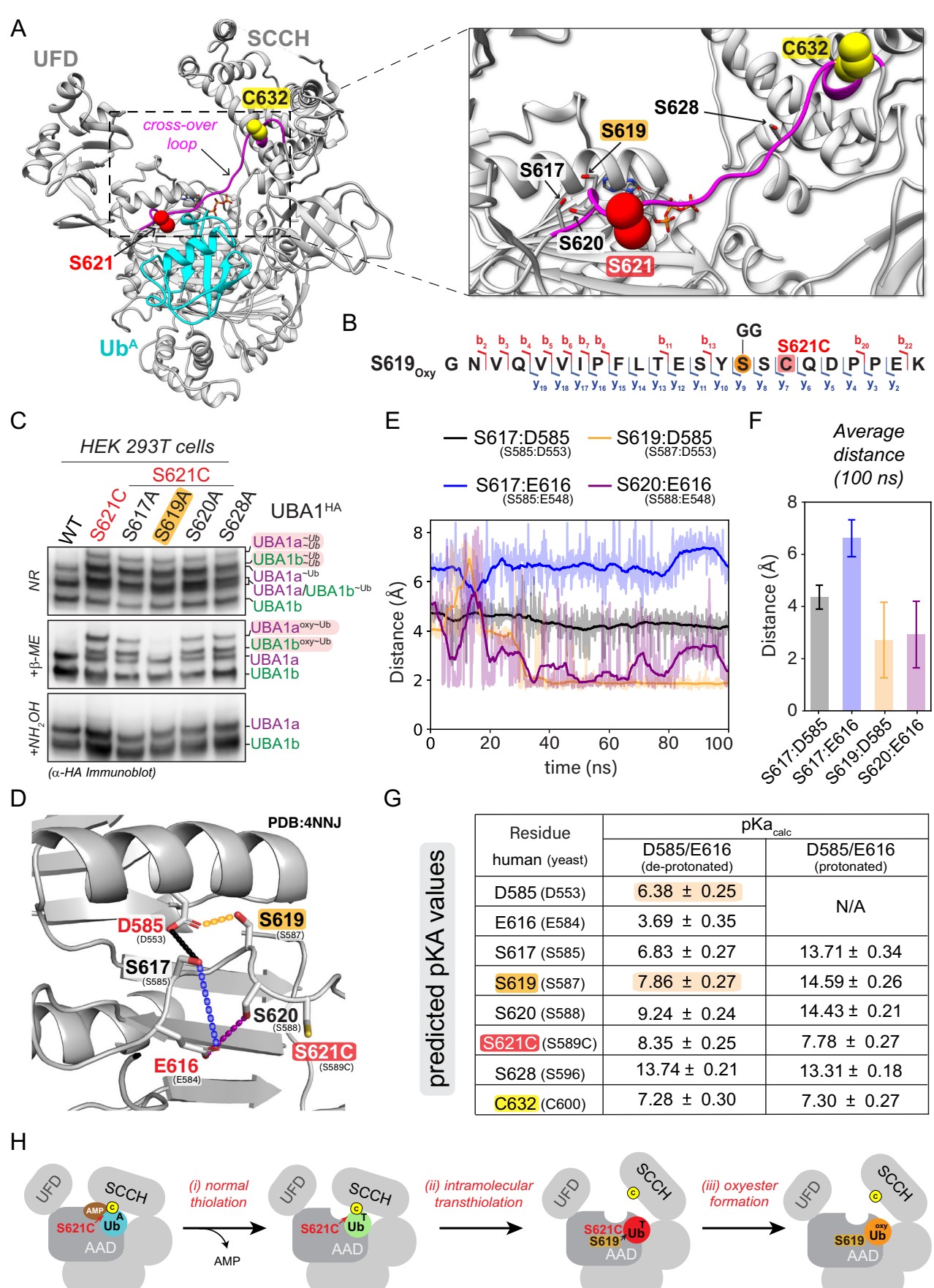

**Figure 4. UBA1 S621C forms an oxyester at S619 via a thioester intermediate.**

(A) Structure of human UBA1 (gray) with ubiquitin (cyan) at the adenylation site (PDB: 6DC6) highlighting S621 (red spheres) and its location in the cross-over loop (magenta). ATP (stick representation with carbon, oxygen, nitrogen, and phosphorus in gray, red, indigo, and orange) was modeled as described for Fig. 2A. An expanded view to the right illustrates that S621 is in the vicinity of the catalytic residue C632 (highlighted in yellow spheres) and in close proximity to four serine residues (shown as sticks with oxygen in red). (B) UBA1 p.S621C forms an oxyester at S619 (highlighted in orange). C-terminally FLAG tagged UBA1 p.S621C was expressed in HEK293T cells, followed by anti-FLAG immunoprecipitation and mass spectrometry analysis. The ubiquitylated residue is colored in orange and labeled with a diGly remnant (GG). Detected b and y ions for the oxyester-containing peptide are highlighted in red and dark blue, respectively. (C) UBA1 p.S621C forms an oxyester at S619 but not at any other nearby serine residue. Indicated UBA1$^{HA}$ variants were expressed in HEK293T cells and lysates were either not treated (non-reducing, NR) or treated with indicated reducing agents and subjected to anti-HA immunoblotting. Additional UBA1 species are highlighted by a red box. (D) Ribbon diagram of the serine-rich region surrounding UBA1 p.S621C following 100 ns of molecular dynamics. The simulation was performed using a yeast UBA1 structure (PDB: 4NNJ) with cysteine substitution of S589 (corresponding to human S621). For consistency, we refer to the corresponding human residues and indicate yeast residues in parentheses. Interactions highlighted with dashed lines are color coded for panels (E) and (F). Sidechain heavy atoms are displayed in gray, red, or yellow for carbon, oxygen and sulfur respectively. Double red sticks indicate carbonyl oxygen atoms. (E) Distance (Å) over time (ns) between serine sidechain hydroxyl hydrogen atoms and the nearest sidechain oxygen of D585 or E616 as indicated and displayed in panel (D). Overlayed with the displayed measurements is a moving window average of 100 ps (solid line). S617 is >4 Å from D585 (black) and >6 Å (blue) from E616. S619 has a stable hydrogen bonding interaction (<1.6 Å) with D585 (orange) and is stable during most of the simulation. S620 forms a transiently stable hydrogen bonding interaction with E616 (purple) fluctuating from 1.6 Å to approximately 3 Å. (F) A plot of the average distance between a serine and one of the two catalytic bases from a 100 ns molecular dynamics simulation. Interactions between S617 with D585 (black) or E616 (blue) are >4 Å, while interactions between S619 with D585 (orange) and S620 with E616 (purple) have an average distance of approximately 3 Å, allowing for a proton transfer to occur. $n = 10001$ frames across the trajectory, error bars = s.d. (G) Calculated pK$_a$ values of the D585 and E616 catalytic base and S617, S619, S620, and S628 sidechain oxygen atoms and of the S621C and the catalytic cysteine C632 sulfur atoms. The pK$_a$ values were also calculated for a structure in which a D585 and E616 sidechain oxygen atom were protonated (D585-H/E616-H). D585, but not E616, has an increased calculated pK$_a$ with a value of 6.38 ± 0.25, which is amenable for its activity as a base at physiological pH. All serines in proximity to D585 and E619 display reduced pK$_a$ values, with S617 and S619 close to physiological pH, making them candidates for deprotonation by D585. When D585 and E619 are protonated, the pK$_a$ values of the serines return to their expected value of approximately 13, suggesting D585 and E619 are driving their changes in the pK$_a$ values. (H) Proposed mechanism of oxyester formation for the UBA1 p.S621C variant. Source data are available online for this figure.

## SMA-associated and a subset of LCINS mutations affect protein folding and render UBA1 activity thermolabile

Previous studies have shown that missense mutations in UBA1 cause spinal muscular atrophy (SMA) (Ramser et al, 2008) and are associated with lung cancer in never smokers (LCINS) (Zhang et al, 2021) (Fig. 6A; Table EV2). We next wondered how these mutations would compare to the non-canonical VEXAS mutations and subjected them to mechanistic profiling. As expected, none of the SMA and LCINS mutations led to UBA1c isoform expression in cells (Appendix Fig. S2A). We noted distinct localizations of different disease-associated mutations in the UBA1 structure. While non-canonical VEXAS mutations cluster close to the binding site for ubiquitin and ATP, LCINS mutations are structurally more dispersed (Fig. 6B). SMA mutations are also concentrated in the AAD, but located further away from the ubiquitin and ATP binding pockets when compared to the non-canonical VEXAS mutations, suggesting a more structural impact. Indeed, several LCINS and SMA mutations were at residues that engage in intramolecular hydrophobic or electrostatic interactions, indicating that they would likely affect UBA1 activity through changes in protein fold and stability. Consistent with this hypothesis, the LCINS-associated p.L904R mutant aggregated during purification from *E. coli* and could only be produced using an abbreviated protocol (Appendix Fig. S2B). In contrast, all other SMA and LCINS mutants could be produced by the standard protocol to high purity (Appendix Fig. S2C). To test whether these mutations would cause more subtle conformational changes with impact on UBA1 activity, we next pre-incubated UBA1 proteins at varying temperatures in the absence of substrate and subsequently subjected them to ubiquitin thioester formation assays on ice (Fig. 6C). Consistent with our previous report (Poulter et al, 2021), we found that, compared to WT UBA1, the p.S56F mutant lost its ability to form thioesters when pre-incubated at lower temperatures, demonstrating that it renders UBA1 activity thermolabile (Fig. 6D; Appendix Fig. S2D).

None of the other non-canonical VEXAS mutations had such an effect. In contrast, all SMA mutations (p.M539I, p.S547G, p.E557V) and two LCINS mutations (p.Q469P, p.D555Y), caused a pre-incubation temperature-dependent decrease in UBA1 thioester formation (Fig. 6E,F; Appendix Fig. S2E,F). Thus, opposed to non-canonical VEXAS mutations, thermolability is a shared feature of SMA mutations and is also present in a subset of LCINS mutations.

## LCINS-associated UBA1 mutations p.Q724P and p.H643Y form aberrant ubiquitin thioesters and exhibit a bottleneck in ubiquitin transfer to E2 enzymes

For the two LCINS mutations that did not exhibit thermolability, UBA1 p.Q724P and p.H643Y, we noted additional ubiquitin species when we performed UBA1 thioester formation assays at 37 °C (Appendix Fig. S3A–C). We found that these aberrant conjugates are consistent in size with 2–3 additionally charged ubiquitins, that they become more prevalent with increasing incubation time, and that their formation is dependent on the catalytic cysteine (Fig. 7A; Appendix Fig. S3D). Moreover, these additional ubiquitin species were mostly reducible by β-ME, except for a minor fraction at 60 min of reaction time, which required NH$_2$OH for reduction. These results suggested that during the thiolation reaction, UBA1 p.Q724P and p.H643Y form multiple aberrant thioesters and, to a lesser extent, oxyesters. To determine the underlying molecular basis, we first focused on UBA1 p.H643Y. Analysis of a previously reported UBA1 structure in its doubly loaded state revealed that H643 normally engages in hydrogen bonds with N639 and a water molecule (Fig. 7B, upper panel) (Schafer et al, 2014). H-to-Y substitution at UBA1 p.643 results in loss of these interactions as well as steric clashes with I629 and T633, which together likely lead to structural rearrangements close to the catalytic cysteine residue (Fig. 7B, lower panel). Similarly, molecular dynamics simulations revealed that the p.Q724P mutation partially disrupts the backbone hydrogen-bond in its own α-helix and kinks the helix, thereby

 

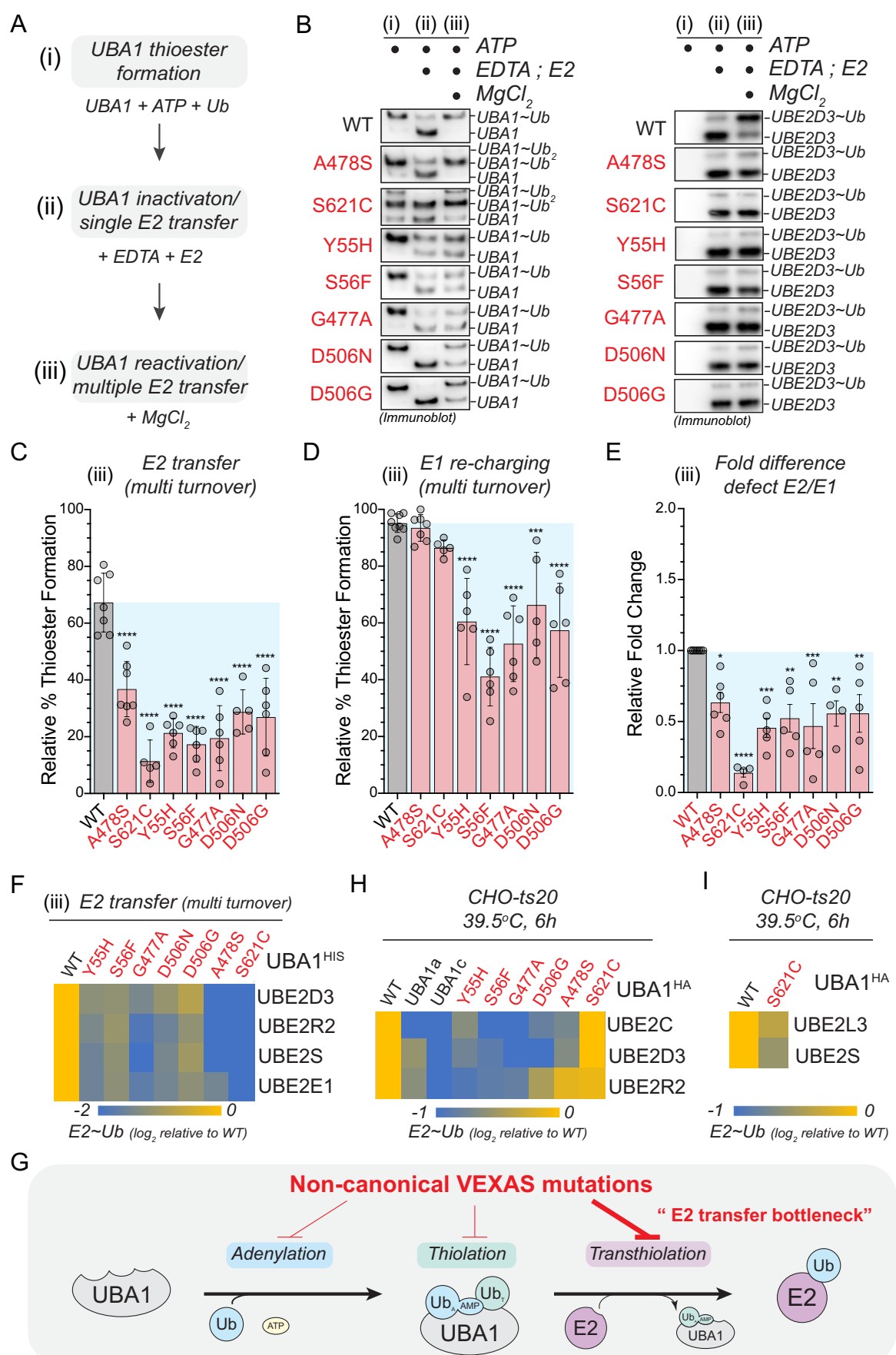

**Figure 5. Most non-canonical VEXAS mutations exhibit a bottleneck in ubiquitin transfer to E2 enzymes.**

(A) Schematic overview of the sequential, three-phase in vitro assay used to measure UBA1 transthiolation. (i) Charging of UBA1 by incubation of 250 nM UBA1b with 10 µM ubiquitin and 5 mM ATP (ii) Quenching of UBA1 charging and single transfer to E2 enzyme by addition of 100 mM EDTA and 1 µM UBE2D3 (iii) Reactivation of UBA1 charging and multi-transfer to E2 enzyme by addition of 100 mM MgCl$_2$. (B) Non-canonical VEXAS mutations (red) are impaired in E2 transthiolation in vitro. Immunoblot analysis of the sequential, three-phase in vitro assay described in panel (A) using antibodies against UBA1 (left panel) or UBE2D3 (right panel). (C) Non-canonical VEXAS mutations (red) are impaired in E2 transthiolation in vitro, as revealed by quantifications of relative E2 thioester levels (UBE2D3-Ub/total signal) of multi-turnover reactions (phase iii) depicted in panel (B). $n = 5$–7 biological replicates as indicated, mean $-/+$ s.d., ****$p < 0.0001$, one-way ANOVA. (D) Some non-canonical VEXAS mutations exhibit ~2-fold defects in UBA1 re-charging, as revealed by quantifications of relative UBA1 thioester levels (UBA1-Ub/total signal) of multi-turnover reactions (phase iii) depicted in panel (B). $n = 5$–7 biological replicates as indicated, mean $-/+$ s.d., ***$p < 0.001$, ****$p < 0.0001$, one-way ANOVA. (E) Non-canonical VEXAS mutations most strongly affect the E2 transthiolation step in vitro, as revealed by quantification of the relative E2 transthiolation over the relative UBA1 re-charging defects of multi-turnover reactions (phase iii) depicted in panel (B). $n = 5$–7 biological replicates as indicated, mean $-/+$ s.d., *$p < 0.05$, **$p < 0.01$, ***$p < 0.001$, ****$p < 0.0001$, one-way ANOVA. (F) Non-canonical VEXAS mutations (red) are defective in charging different classes of E2s in vitro. Heatmap depicting the relative log$_2$-fold changes in E2 ubiquitin thioester formation measured by multi-turnover reactions (phase iii) using FITC-labeled ubiquitin (for details refer to Fig. EV4A–C). For quantifications, E2 ubiquitin thioester levels in phase (iii) were first normalized to the total FITC signal in phase (i) and then normalized to WT. $n = 3$ biological replicates per condition. (G) CHO cells with UBA1a, UBA1c, or non-canonical VEXAS mutants (red) as sole source of UBA1 generally exhibit lower ubiquitin thioester levels for select E2 enzymes as compared to CHO cells with WT UBA1. CHO rescue assays were performed as described in Fig. 1D, followed by anti-E2 immunoblotting. Heatmap depicts the relative log$_2$-fold changes in ubiquitin thioester levels of UBE2C, UBE2D3, and UBE2R2. $n = 3$ biological replicates per condition. (H) CHO cells with the p.S621C mutant as sole source of UBA1 exhibit lower ubiquitin thioester levels for UBE2S and UBE2L3 as compared to CHO cells with WT UBA1. CHO rescue assays were performed as described in Fig. 1D, followed by anti-UBE2L3 and anti-UBE2S-immunoblotting and quantification. $n = 3$ biological replicates per condition. (I) Schematic representation of our in vitro findings, revealing that non-canonical VEXAS mutations inactivate UBA1 by most prominently affecting ubiquitin transfer to E2 enzymes. Source data are available online for this figure.

forcing structural rearrangements of the helix containing the catalytic cysteine residue and the thioester-bound ubiquitin (Fig. 7C; Appendix Fig. S3E). Together, these findings suggest that both, p.Q724P and p.H643Y, reconfigure UBA1's catalytic thiolation center. We hypothesize that these structural changes render the ubiquitin thioester more exposed to intramolecular attack by other cysteine residues during thiolation, resulting in the formation of aberrant thioesters (Fig. 7D). As with the oxyester-forming non-canonical VEXAS mutations UBA1 p.A478S and p.S621C (Fig. 5A–E), we found that the major molecular defect of UBA1 p.Q724P and p.H643Y is ubiquitin transfer to E2 enzymes, as evidenced by quantifications of multiple turnover reactions that showed a ~ 3-fold reduction in ubiquitin transfer to UBE2D3, while UBA1 re-charging was largely unaffected (Fig. 7E,F). We made comparable findings for UBE2R2, UBE2S, UBE2L3, and UBE2E1 (Appendix Fig. S3F–H). Thus, similarly to non-canonical VEXAS mutations, LCINS-associated UBA1 p.Q724P and p.H643Y form aberrant thioesters and bottleneck at the E2 transthiolation step.

## Thermolabile SMA and LCINS mutations have less severe defects than other UBA1 mutations

During our systematic profiling approach we noted that in contrast to the folding-deficient p.L904R and the multi-thioester-forming p.H643Y and p.Q724P mutants, the thermolabile LCINS mutations (p.Q469P, p.D555Y) did not exhibit marked defects in ubiquitin adenylation, thiolation, or transthiolation at our chosen in vitro assay conditions (Fig. 7E,F; Appendix Figs. S3A–C,F–H and S4A,B). This was also the case for the thermolabile SMA mutations (Appendix Fig. S4C,K). Consistent with these in vitro findings, thermolabile LCINS and SMA mutations (p.Q469P, p.D555Y, p.M539I, p.S547G, p.E557V) showed no obvious defects in charging of select E2s or polyubiquitylation in the CHO cell rescue model (Fig. 7G; Appendix Fig. S5A–F). However, LCINS mutations exhibiting the E2 bottleneck (p.Q724P, p.H643Y) were impaired in charging of select E2s and showed reduced polyubiquitylation levels, similar to what we had observed for the non-canonical VEXAS mutations (Figs. 7G, 5H,I and EV4D–G; Appendix

Fig. S5A–F). These results suggest that thermolabile SMA and LCINS mutations confer less severe cellular defects than UBA1 mutations.

Taken together, our systematic mutation profiling approach revealed that disease-associated UBA1 missense mutations can be generally categorized into two classes. First, mutations that most prominently affect the E2 transthiolation step in vitro and cause defects in E2 charging and polyubiquitylation in cells; second, mutations that primarily render UBA1 activity thermolabile in vitro and cause only subtle defects in our cellular model system (Fig. 8). While LCINS mutations comprise both classes, SMA mutations are exclusively thermolabile and non-canonical VEXAS mutations bottleneck at the E2 transfer step (most prominently by affecting transthiolation but also by additional defects at earlier stages of ubiquitin activation). From these observations, we infer distinct and shared molecular mechanisms of UBA1 inactivation across different disease states.

## Discussion

UBA1 initiates the vast majority of cellular ubiquitylation and plays an essential role in most, if not all, cellular pathways (Jin et al, 2007; McGrath et al, 1991; Schulman and Harper, 2009). Given the pleiotropic role of this enzyme in biologic processes, it is surprising that disease-causing mutations in *UBA1* can result in highly specific clinical manifestations, ranging from severe inflammation in VEXAS syndrome to lung cancer in never smokers (LCINS) and X-linked spinal muscular atrophy (SMA). How these different missense mutations impact UBA1 and lead to specific phenotypes remains unclear. In this work, through mechanistic profiling of UBA1 variants, we identify different classes of mutations that indicate common and distinct features of UBA1 inactivation across these different disease states (Fig. 8). These findings have important implications for mechanisms of ubiquitin activation during normal and pathophysiological processes and lay the foundation for developing novel therapeutic strategies for VEXAS syndrome and other *UBA1*-related diseases in the future.

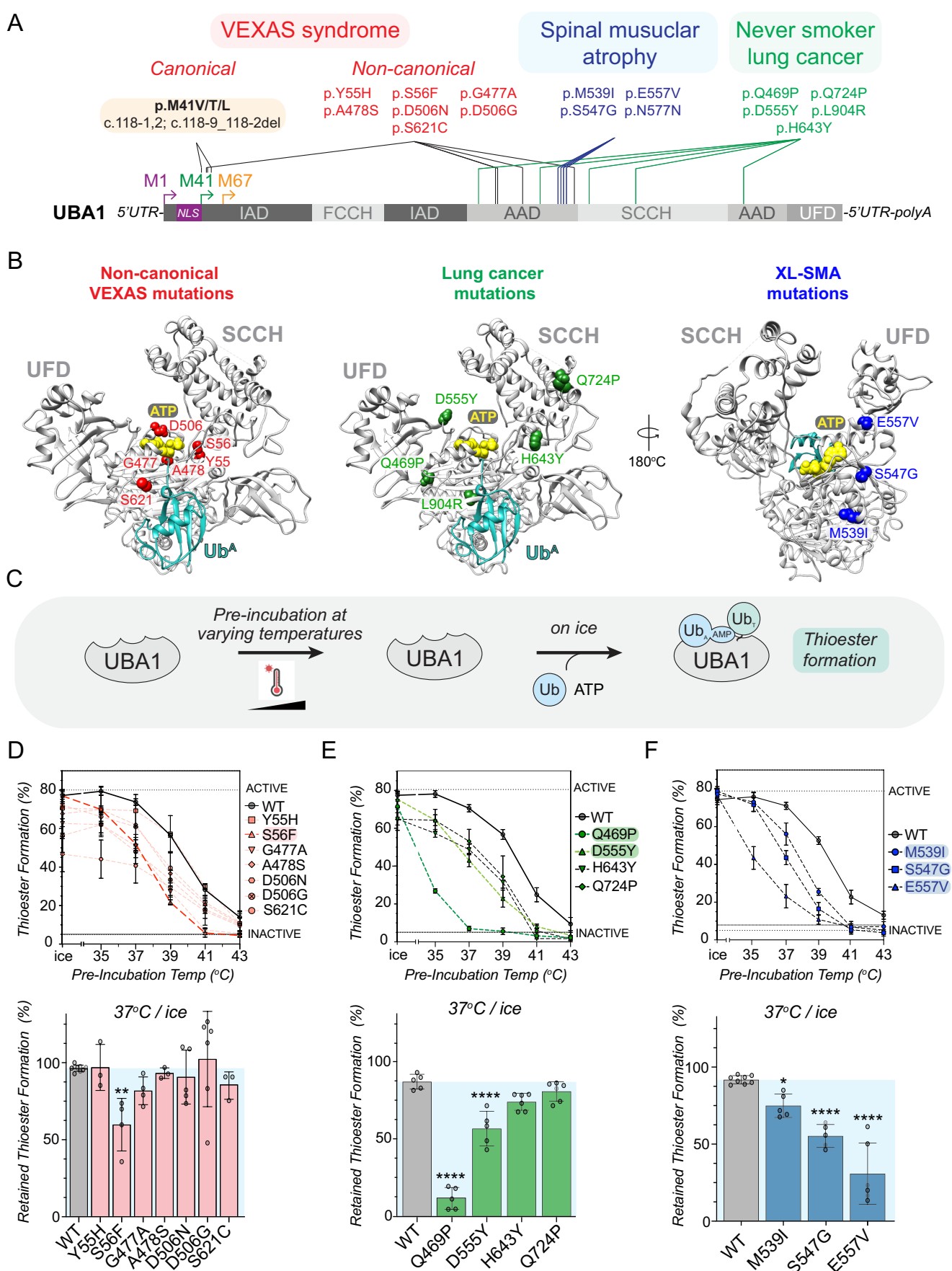

**Figure 6. SMA-associated UBA1 mutations and a subset of never smoker lung cancer mutations render UBA1 activity thermolabile.**

(**A**) Schematic overview of UBA1 domains highlighting the location of mutations causing canonical and non-canonical VEXAS syndrome (red), causing spinal muscular atrophy (SMA, blue), or implicated in lung cancer in never smokers (LCINS, green). (**B**) Structural mapping of the position of the different disease-associated UBA1 amino acid substitutions using the crystal structure of human UBA1 (gray) with ubiquitin (cyan) in the adenylation site (PDB: 6DC6) and ATP (yellow spheres) modeling as described for Fig. 2A. Non-canonical VEXAS mutations (red) are clustered around the ATP binding site. SMA mutations (blue) are also in the AAD but on the opposite side of the catalytic adenylation center. LCINS mutations (green) are structurally more dispersed and present in both the AAD and SCCH domain. (**C**) Overview of the experimental workflow to test for a temperature-dependent impact of mutations on UBA1 activity. UBA1 proteins were pre-incubated at varying temperatures in the absence of substrate and subsequently subjected to ubiquitin thioester formation assays on ice using 500 nM UBA1b, 10 µM ubiquitin, and 5 mM ATP. (**D**) UBA1 p.S56F is the only non-canonical VEXAS mutation that renders UBA1 thiolation thermolabile. Indicated UBA1 proteins were subjected to the assay described in panel (**C**) followed by anti-UBA1 immunoblotting. *Top graph*: relative UBA1 thioester levels (UBA1-Ub/total) were quantified and plotted against the pre-incubation temperatures. Compared to WT UBA1, only the p.S56F variant is more greatly reduced in activity with increasing pre-incubation temperatures. *Bottom graph*: Quantification of the relative percentage of UBA1-Ub thioester retained after 37 °C pre-incubation as compared to pre-incubation on ice. $n = 3-9$ biological replicates as indicated, mean $-/+$ s.d., $**p < 0.01$, one-way ANOVA. (**E**) LCINS mutations p.Q649P and p.D555Y render UBA1 thiolation thermolabile. Indicated UBA1 proteins were subjected to the assay described in panel (**C**) followed by anti-UBA1 immunoblotting. *Top graph*: relative UBA1 thioester levels (UBA1-Ub/total) were quantified and plotted against the pre-incubation temperatures. *Bottom graph*: Quantification of the relative percentage of UBA1-Ub thioester retained after 37 °C pre-incubation as compared to pre-incubation on ice. $n = 5$ biological replicates for each condition, mean $-/+$ s.d., $****p < 0.0001$, one-way ANOVA. (**F**) All SMA-associated mutations render UBA1 thiolation thermolabile. Indicated UBA1 proteins were subjected to the assay described in panel (**C**) followed by anti-UBA1 immunoblotting. *Top graph*: relative UBA1 thioester levels (UBA1-Ub/total) were quantified and plotted against the pre-incubation temperatures. *Bottom graph*: Quantification of the relative percentage of UBA1-Ub thioester retained after 37 °C pre-incubation as compared to pre-incubation on ice. $n = 5-9$ biological replicates as indicated, mean $-/+$ s.d., $*p < 0.05$, $****p < 0.0001$, one-way ANOVA. Source data are available online for this figure.

Previous biophysical and structural studies have provided a detailed understanding of the concerted intramolecular changes that UBA1 must undergo to ensure ubiquitin activation and conjugation to E2 enzymes (Cappadocia and Lima, 2018). In our work, we determine how mutations implicated in diverse human diseases can interfere with these processes. Particularly, in our in vitro assays we find that single point mutations in the cross-over loop, in the adenylation site, or in proximity of the catalytic cysteine residue result in UBA1 auto-ubiquitylation via aberrant thio- or oxyesters to ultimately inhibit ubiquitin transthiolation to E2 enzymes (Figs. 3–5 and 7). Our results thus highlight the exquisite precision of conformational changes required for faithful ubiquitin activation that when perturbed by subtle changes causes human diseases. In this context, given that UBA1 p.A478S and p.S621C form stable oxyesters in cells (Figs. 1E and 3A), it will be interesting to explore whether in addition to interfering with transthiolation, these mutations might also lead to aberrant recruitment of ubiquitin effector proteins or sequestration of E2 enzymes. Such additional cellular effects of UBA1 p.A478S and p.S621C might explain the apparent discrepancy of their severe in vitro transthiolation defects and their less pronounced influence on E2 charging despite significantly reduced polyubiquitylation levels in the CHO rescue model (Figs. 5F–I and 7G). Moreover, such putative E2-trapping properties of these oxyester-forming variants could be exploited to study cellular E1-E2 interactions, a largely unexplored research area.

Our identification and mechanistic dissection of novel VEXAS-causing mutations expand the understanding of the underlying molecular disease mechanisms. Canonical p.Met41 and splice site mutations inactivate cytoplasmic UBA1 activity through a translational isoform swap, while non-canonical VEXAS mutations inactivate catalytic activity of nuclear and cytoplasmic UBA1 (Figs. 1–4). Our study therefore demonstrates that UBA1c is not required to elicit disease phenotypes. Non-canonical VEXAS mutations reduce UBA1 activity by diverse mechanisms, including reduction in ubiquitin adenylation and thioester formation as well as aberrant formation of oxyesters. For these non-canonical VEXAS mutants, the E2 transfer step is most strongly affected, creating a

bottleneck (Fig. 5). We thus propose that the major molecular mechanism underlying canonical and non-canonical VEXAS syndrome is loss of ubiquitin conjugation of cytoplasmic E2 enzymes. In this work, we did not characterize both known or novel non-coding or splice site *UBA1* mutations, which are emerging as additional causes of disease.

The E2 bottleneck found in non-canonical VEXAS and some LCINS mutations and the correlation between in vitro and cellular defects suggest shared but distinct mechanisms of disease. Ubiquitin charging of all measured E2s is similarly affected in our in vitro system (Figs. 5A–F, 7E, F and EV4A–C; Appendix Fig. S3F–H), suggesting that there is no structural specificity towards particular E2s conferred by these mutations. However, in cells using a CHO reconstitution system, E2-specific defects can be revealed (Figs. 7G and EV4D–G; Appendix Fig. S5), potentially suggesting tissue-specific low abundance or dose-sensitive E2s determining distinct tissue phenotypes. By this logic, low abundance E2s and/or E2s with the most critical functions in the cytoplasm would be most dramatically affected in VEXAS syndrome. Identifying these E2s in future studies could reveal the specific pathways involved in disease pathogenesis, and could explain the recently reported aberrant inflammation of hemato-poietic stem cells of VEXAS patients and their differentiation bias towards myeloid cells (Wu et al, 2023). For LCINS and SMA caused by thermolabile mutations with mild cellular phenotypes (Figs. 6 and 7G), these may represent tissue-specific, highly dose-sensitive E2s. Similar paradigms have been established for ribosomopathies, in which hypomorphic alleles in ribosome components and the associated biogenesis machinery lead to tissue-specific diseases by reducing the translation of the most critical transcripts required for a particular physiological process (Farley-Barnes et al, 2019; Genuth and Barna, 2018; Mills and Green, 2017). We propose that akin to ribosomopathies, disease-specific *UBA1* variants expose the most critical E2s and roles of ubiquitylation in a particular tissue or differentiation stage. Identifying such specific E1-E2 modules by leveraging particular UBA1 mutations in future mechanistic studies will be an exciting area of future investigation. In addition, as suggested by recent reports of UBA1 p.S56F patients exhibiting

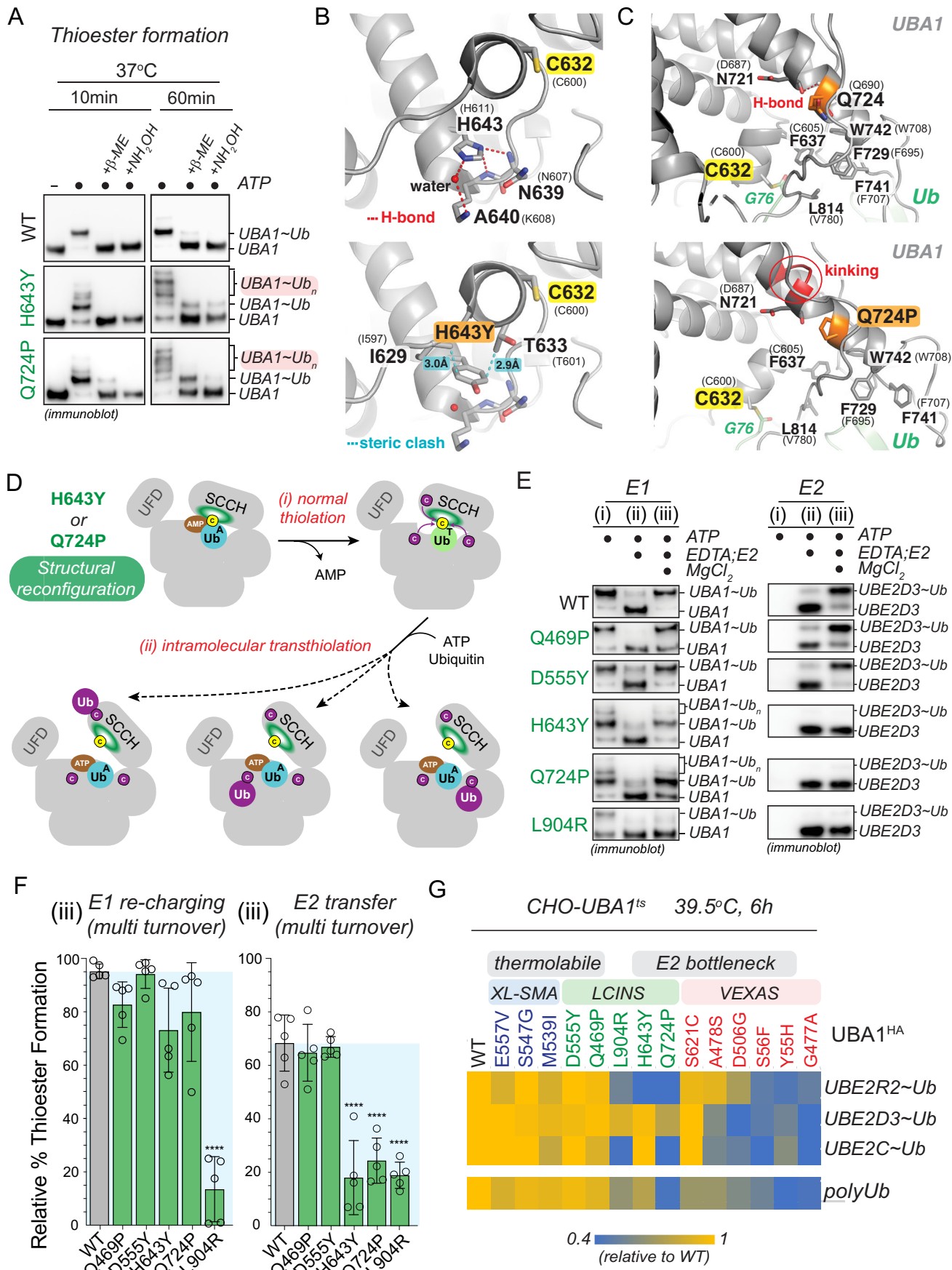

◀ **Figure 7. LCINS mutations UBA1 p.H643Y and p.Q724P form aberrant ubiquitin thioesters and exhibit an E2 transfer bottleneck.**

(A) UBA1 p.H643Y and p.Q724P (green) form aberrant ubiquitin thioester species in vitro. Denoted recombinant UBA1b proteins (500 nM) were incubated with 10 μM ubiquitin and 5 mM ATP for 10 or 60 min at 37 °C. Reactions were treated with reducing agents (β−ME or NH₂OH) as indicated, followed by anti-UBA1 immunoblot analysis. (B) Ribbon diagram of an expanded region of *S. cerevisiae* UBA1 (PDB: 4NNJ) wild-type protein (upper panel) and with H643Y substitution (lower panel). Labeling for this panel and panel (C) follows Fig. 3D. The mutation disrupts hydrogen bonds (red dashed lines) from H643 to N639 and a bound water molecule (red sphere) and forces rearrangements to prevent steric clashes between Y643 and T633/I629 (cyan dashed lines). Stick representation is used to display sidechain heavy atoms of UBA1 residue I629, C632, T633, N639, A640, and H643. In this panel and panel (C), oxygen, nitrogen, and sulfur are colored red, blue, and yellow, respectively. (C) Ribbon diagram of an expanded region of *S. cerevisiae* UBA1 (gray, PDB: 4NNJ) wild-type protein (upper panel) with a thioester-linked ubiquitin (light green) and the final frame of a 100 ns molecular dynamics simulation of UBA1 Q724P mutant (lower panel). A hydrogen bond (red dashed line) between the N721 backbone oxygen atom and Q724 (orange) backbone nitrogen atom is disrupted in the Q724P mutant by kinking of the helix (noted in red). The thioester bond between UBA1 C632 and ubiquitin G76, as well as the sidechain heavy atoms of UBA1 residue F637, N721, Q724, F729, F741, W742 and L814 are displayed by stick representation. (D) Model of how UBA1 p.H643Y and p.Q724P lead to the formation of aberrant UBA1 ubiquitin thioesters via intramolecular transthiolation reactions. (E) Aberrant thioester-forming (p.H643Y, p.Q724P) but not thermolabile (p.Q469P, p.D555Y) LCINS mutations (green) are impaired in E2 transthiolation in vitro. Immunoblot analysis of the sequential, three-phase in vitro assay described in panel 5A using antibodies against UBA1 (left panel) or UBE2D3 (right panel). (F) Quantifications of relative UBA1 re-charging (UBA1-Ub/total signal) and relative E2 thioester levels (UBE2D3-Ub/total signal) of multi-turnover reactions (phase iii) depicted in panel (E). UBA1 p.H643Y and p.Q724P significantly reduce transthiolation while not markedly affecting thiolation, revealing an E2 bottleneck. $n = 5$ biological replicates, mean $-/+$ s.d., ****$p < 0.0001$, one-way ANOVA. (G) CHO cells with E2 bottleneck but not with thermolabile disease mutants as sole source of UBA1 exhibit markedly lower ubiquitin thioester levels for select E2 enzymes and reduced polyubiquitylation as compared to CHO cells with WT UBA1. CHO rescue assays were performed as described in Fig. 1D, followed by anti-E2 and anti-ubiquitin immunoblotting. Heatmap depicts the relative $\log_2$-fold changes in polyubiquitylation or ubiquitin thioester levels of UBE2C, UBE2D3, and UBE2R2. $n = 3$ biological replicates per condition. Non-canonical VEXAS mutations (red), LCINS mutations (green), SMA mutations (blue). Source data are available online for this figure.

predominantly hematological manifestations (Al-Hakim et al, 2023), it will be interesting to see whether there are differences in disease severity between carriers of canonical and non-canonical VEXAS mutations or between LCINS patients with E2 bottleneck and thermolabile variants, once more individuals are identified. Further work is required to determine if specific clinical features, lack of vacuoles in non-canonical *UBA1* VEXAS cases, or severity of inflammatory pathway activation is differentially regulated by distinct *UBA1* mutations.

Our cross comparison between *UBA1*-related diseases should consider distinct features of each of these phenotypes and associated genetics. UBA1 is an X-chromosomally encoded gene thought to escape X-chromosomal inactivation (Tukiainen et al, 2017), is essential in lower eukaryotes, and is intolerant to loss of function mutations in healthy individuals (Gudmundsson et al, 2021). For VEXAS syndrome, somatic mutations in males cause disease, leading to the inferences that both heterozygous mutations in euploid females are unlikely to lead to inflammation and that germline mutations are likely embryonic lethal. For SMA, disease-causing mutations have been identified as germline in hemizygous males. The fact that SMA mutations are thermolabile and have minute to undetectable defects in our cellular model system (Figs. 6 and 7G) may be consistent with severe variants being embryonic lethal and permitting survival for mild mutations only. The lack of associated inflammation in patients with SMA may be masked by other systemic effects of UBA1 loss or not present due to the mild effects of mutations on UBA1 function. Finally, for LCINS, these mutations are again somatic but found exclusively in females, and many are non-sense or frameshift in addition to missense mutations, suggesting a distinct genotype (heterozygous) and tissue distribution associated with a unique disease mechanism. It would be intriguing to determine whether UBA1 mutations found enriched within specific diseases in the somatic state (i.e., VEXAS, LCINS), would remain pathogenic when present in the germline setting or when present in other unexplored tissues. Together, our results highlight similarities and distinct attributes associated with these disease subtypes and may reveal potential risk stratification of

variants across different diseases, overlapping treatment approaches, and shared mechanisms of disease.

## Methods

### Human subjects

The Ethical Review Boards of NYU (NCT06004349) approved the study. All investigations were performed in accordance with the ethical standards of the 1964 Declaration of Helsinki and its later amendments. Consent was obtained from all subjects for the performed experiments and for publishing the patient photos in Fig. EV1.

### Exome sequencing and analysis

Genomic DNA samples for exome sequencing were prepared from peripheral blood using the Qiagen DNA Easy kit. Exome sequence libraries were prepared using an Illumina TruSeq DNA Sample Preparation Kit version 2. Paired-end sequencing was performed on the Illumina HiSeq2000 instrument. BWA software was used to align the sequence reads to the Human Reference Genome Build hg19. Picard was used to identify PCR duplicates. GATK was used to realign the sequence reads around microindels to produce a better alignment and to recalibrate the base qualities to obtain more accurate quality scores. GATK Unified Genotyper and SAMtools were then used to identify SNVs and Indels. Mutect2 was used for somatic calling. ANNOVAR was used for annotation and VCF files were generated.

### Digital droplet PCR

We analyzed somatic *UBA1* mutations using DNA extracted from peripheral blood or sorted cell populations. Specific probes were generated for *UBA1* variants. Reactions were performed using 11 μL 2x ddPCR Supermix for probes, 900 nM target-specific PCR primers, and 250 nM mutant-specific (FAM) and wild-type-specific

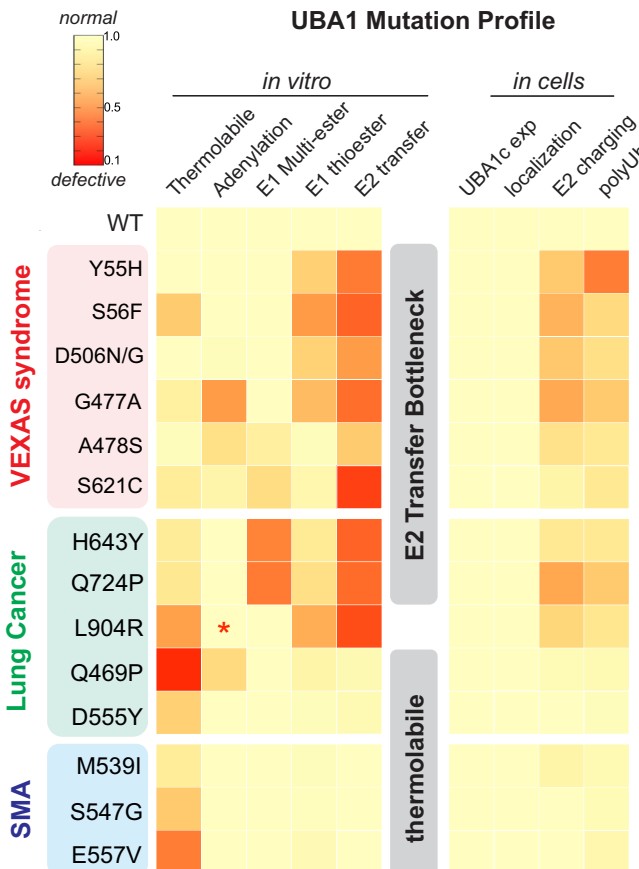

**Figure 8. Systematic mutation profiling defines shared and distinct mechanisms of UBA1 inactivation in different human diseases.**

Heatmap summarizing the findings of the of biochemical and cellular characterization of disease-associated UBA1 mutations conducted in this study. For each determined parameter, defects were normalized to WT, which was set to 1. We find that UBA1 mutations fall into two classes that (i) bottleneck at the E2 transfer step in vitro and reduce levels of E2 ubiquitin thioesters and polyubiquitylation when present as sole source of E1 in CHO cells and (ii) render UBA1 activity thermolabile in vitro and only show small cellular defects. While LCINS mutations (green) are comprised of both classes, SMA mutations (blue) are exclusively thermolabile and non-canonical VEXAS mutations (red) bottleneck at the E2 transfer step, suggesting distinct and shared molecular mechanisms of UBA1 inactivation across different disease states. * = not determined. Source data are available online for this figure.

(HEX) probes. 20 µL of PCR mixture and 70 µL Droplet generation oil were mixed, and droplet generation was performed using a Bio-Rad QX100 Droplet Generator. The droplet emulsion was thermally cycled in the following conditions: denaturing at 95 °C for 10 min, 40 cycles of PCR at 94 °C for 30 s and at 55 °C for 1 min, and a final extension at 98 °C for 10 min. PCR amplification in the droplets was confirmed using Bio-Rad QX200 Droplet Reader. The threshold was determined by comparing the non-template ddPCR results. All the data were evaluated above the threshold. QuantaSoft (Biorad) was used to analyze the % variant allele fraction data.

## Cell lines, media, and transfections

Parental Chinese hamster ovary (CHO) cell line (e36) and temperature sensitive UBA1 knockdown CHO cell line (ts20) (Lenk et al, 1992)

were cultured in complete CHO ts20 medium (MEM α [Gibco, 12571063] supplemented with 1.8 g/L glucose, 10% FBS, and Penicillin-Streptomycin 100 U/mL) and maintained at 30.5 °C with 5% carbon dioxide. HEK293T (ATCC, CRL-3216) and HeLa cells (ATCC, CCL-2) were cultured in DMEM (Gibco, 11966025) supplemented with 10% FBS, and Penicillin-Streptomycin 100 U/mL and maintained at 37 °C with 5% carbon dioxide. All plasmid transfections were performed using PEI in a 1:5 DNA-to-PEI ratio. Cell lines were obtained from the indicated sources, expanded, and not further authenticated. Cells were routinely tested for mycoplasma contamination using the MycoAlert Mycoplasma detection kit from Lonza (LT07-118).

## Lentiviral generation

To generate lentiviral particles, HEK293T cells were transfected with pMD2.G (Addgene #12259), psPAX2 (Addgene #12260), and pHAGE packaging vector containing UBA1 variants in a DNA mass ratio of 2:3:4, respectively, using Lipofectamine 2000 (Invitrogen). Supernatant was harvested 72 h post-transfection, filtered through a 45 µm syringe-driven filter, and purified overnight using Lenti-X Concentrator (Takara Bio). Concentrated lentiviral pellets were then resuspended in complete CHO ts20 medium and either used immediately or snap-frozen in liquid nitrogen and stored at −80 °C.

## In vivo UBA1 characterization using CHO ts20

Characterization of UBA1 variants in CHO ts20 cell lines was conducted as previously described (Beck et al, 2023). Briefly, UBA1 variant cell lines were generated by lentiviral transduction with 1.6 µg/mL polybrene using lentiviral particles containing HA-FLAG tagged WT UBA1, single isoform UBA1 (UBA1a and UBA1c), VEXAS Syndrome UBA1 variants (Y55H, S56F, G477A, D506G, A478S, and S621C, and M41V), LCINS-associated UBA1 variants (Q469P, D555Y. H643Y, Q724P, and L904R), or SMA-associated UBA1 variants (M539I, S547G, and E557V). After 24 h, lentiviral-containing media was removed and replaced with fresh complete CHO ts20 medium. Transduced ts20 lines were allowed to grow for another 48 h before supplementing media with 5 µg/mL puromycin.

UBA1 variant CHO ts20 cell lines were harvested by trypsinization, counted, pelleted at 300 × g for 10 min, media aspirated, and pellet resuspended in complete CHO ts20 medium. Resuspended cells were normalized to one million cells per mL of media. Resuspended cells were then transferred to 1.5 mL microcentrifuge tubes and moved to pre-warmed thermomixers set to 39.5 °C. Samples were then incubated shaking at 500 rpm for 6 h. Following incubation, heat treated samples were pelleted at 1200 × g for 10 min, supernatant aspirated, and pellets resuspended in Urea-SDS lysis buffer. Samples were then sonicated for 15 s at 30% amplitude. Sonicated samples were heated at 65 °C for 5 min. Finally, whole cell lysates were separated by sodium dodecyl sulfate (SDS) polyacrylamide gel electrophoresis and analyzed by immunoblotting. Immunoblotting blocking was performed with 3% skim milk.

## Proteins, antibodies, and other reagents

Ubiquitin, Fluorescein isothiocyanate (FITC)-labeled ubiquitin, and ATP were purchased from ThermoFisher. UBE2D3, UBE2R2, UBE2S, UBE2L3, and UBE2E1 were purchased from R&D Systems.

Primary antibodies for UBA1a/b (Cell Signaling, 4891S), Poly-ubiquitin (Cell Signaling, 3936S), UBE2C (Proteintech, 66087-1-Ig), UBE2D3 (Cell Signaling, 4330S), and UBE2R2 (Proteintech,14077-1-AP), and β-actin (Cell Signaling, 4970) were used at a concentration of 1:1000 and visualized using HRP-conjugated secondary antibodies (anti-rabbit [Cell Signaling, 7074S] or anti-mouse [Cell Signaling, 7076S]) at a concentration of 1:1000. HRP-conjugated antibodies were visualized with Immobilon Western Chemiluminescent HRP Substrate (Millipore, WBKLS0500).

## Cloning of UBA1 expression constructs

For expression in cells, human full-length UBA1 was cloned into a pHAGE-C-FLAG-HA-IRES-PURO (Harper lab) via gateway cloning. For recombinant expression in *E. coli*, UBA1b (UBA1-41-1058) were cloned into pET-28a+ via Gibson assembly (NEB, E5510S). All disease variants were introduced using a Q5 site-directed mutagenesis kit (NEB, E0554S).

## Immunoprecipitations for mass spectrometry analysis

For co-immunoprecipitation experiments, HEK293T transiently were transiently transfected with indicated pHAGE-UBA1-Flag-HA constructs for 48 h. For each condition, typically $2 \times 15$-cm dishes were used. Cells were harvested by scraping, washed in PBS, and centrifuged at $300 \times g$ for 5 min. The cell pellets were either stored at $-80\,^{\circ}\text{C}$ or directly used for immunoprecipitation experiments.

Cells were resuspended in two pellet volumes of ice-cold TB buffer (20 mM HEPES [pH 7.3], 110 mM potassium acetate, 2 mM magnesium acetate, 50 mM NaCl, 1 mM EGTA, 0.1% NP-40, protease inhibitors (Roche)) and incubated on ice for 30 min. For lysis, cells were either sonicated or repeatedly snap-frozen in liquid N2 and thawed, then further sheared by repeated passes through a 25G needle (BD Tuberculin). To remove residual lipids, the supernatant was filtered through a 0.22-µm filter (Millex-GV) and the lysates were cleared by centrifugation at $4\,^{\circ}\text{C}$ at $15,000 \times g$ for 30 min.

The lysate was pre-cleared with ~20 µL Protein G agarose (Sigma) per 1 mL of lysate at $4\,^{\circ}\text{C}$ for 30 min. Subsequently, lysates were incubated with ~15 µL of anti-Flag M2 agarose (Sigma) per 1 mL of lysate for 2–4 h at $4\,^{\circ}\text{C}$. Beads were then washed three times with lysis buffer and eluted in TB buffer supplemented with 0.5 mg/mL 3xFlag-peptide (SIGMA F4799-4MG). Eluted proteins were precipitated by adding 20% TCA followed by overnight incubation on ice. Protein pellets were washed three times with ice-cold 90% acetone in 0.01 M HCl, air dried, and further processed for mass spectrometry analysis as described below.

## Mass spectrometry analysis

Eluates from FLAG IPs were precipitated with TCA overnight, reduced, alkylated, separated from FLAG peptide via S-Trap™ mini columns (Protifi) and in-column digested with trypsin overnight. Tryptic digests were analyzed using an orbitrap Fusion Lumos tribrid mass spectrometer interfaced to an UltiMate3000 RSLC nano HPLC system (Thermo Scientific) using data-dependent acquisition. Specifically, peptides were loaded with autosampler and trapped in an Acclaim PepMap 100 trap column (75 µm × 2 cm, Thermo Scientific) at 4 µL/min with 100% solvent A (0.1% formic acid in water) for 5 min. Peptides were then separated with an Acclaim PepMap RSLC column (75 µm × 25 cm, C18 2 µm 100 Å, Thermo Scientific) at 0.25 µL/min with a gradient of 4–31% solvent B (80% I, 0.1% formic acid) in 90 min followed by a 30 min gradient to 50% B. The column was then washed with 99%B for 5 min before returning to 1% B. Column was equilibrated at 1%B for 10 min before the next injection. Column was maintained at room temperature. Mass spectrometry data was recorded between 8–145 min using data-dependent acquisition with a cycle time of 1 s. Full scan MS1 was acquired in the orbitrap with resolution 240,000 ($m/z$ 200). HCD MS/MS spectra was acquired in the linear ion trap at unit mass resolution with isolation window 1.2 $m/z$ using turbo scan. HCD energy was 30%. AGC was 250% for MS1 and 150% for MS2. Precursors with charges between 2–7 was selected for MS2, Dynamic exclusion was set a ± 10 ppm for 60 sec with isotopes excluded. Initial protein identification was carried out using Proteome Discoverer (V2.5) software (Thermo Scientific) against the human protein database downloaded from Uniprot.org (v2022.06.14.) along with a customer protein database containing UBA1 variants and a common contaminant database. Search parameter include: full trypsin digestion with up to 2 missed cleavages; precursor mass tolerance 20 ppm; fragment mass tolerance 0.5 Da. Fixed modification: methylthio of cysteine (+ 45.988); variable modifications: oxidation of M, ubiquitylation (GG) of K, S, T and protein N-terminal; and protein N-terminal acetylationPercolator FDR control was set at 1% for both protein and peptide ID with concatenated target/decoy database. Proteins with 1 peptide ID were included in the report. Protein and peptide quantification were based on precursor intensity. MS/MS spectra of oxyesters at ubiquitylated S and T residues were manually validated.

## Cellular expression in HEK293T

To analyze cellular isoform expression and UBA1 ubiquitin thioester formation, HEK293T cells were transiently transfected with indicated pHAGE-UBA1-FLAG-HA constructs. Cells were harvested 48 h after transfection, washed with PBS, lysed in urea sample buffer (150 mM Tris [pH 6.5], 6 M urea, 6% SDS, 25% glycerol, 0.01% bromophenol blue), and sonicated. Where indicated, lysates were treated with reducing agents ($NH_2OH$ or β-ME) prior to anti-UBA1 immunoblot analysis.

## Immunofluorescence analysis

To visualize UBA1 subcellular localization, HEK293T cells were transiently transfected with indicated pHAGE-UBA1-Flag-HA constructs. Cells were fixed with 4% formaldehyde in PBS for 20 min, permeabilized with 0.5% Triton in PBS for 10 min, blocked in 2% BSA for 1 h and stained with primary anti-UBA1a/b antibodies (Cell Signaling, 4891S, 1:500) for 1 h. After washing with PBS, cells were incubated with Alexa488-labeled donkey-anti-mouse IgGs (Jackson, Cat#:715-545-150, 1:200), rhodamine-labeled phalloidin (ThermoFisher, R415, 1:1000) and Hoechst 33342 for 1 h. Cells were mounted on slides with SlowFade Gold (Invitrogen). Images were taken using a Nikon A1R + HD confocal microscope system (Nikon Instruments, Melville NY). 488 nm, 561 nm, and 640 nm laser lines provided illumination foechsthst, AF 488, Rhodamine Red X, and AF647 fluorophores, respectively. Data were acquired using Galvano mode at

$1024 \times 1024$ with no line averaging A Z-piezo stage (Physik Instrumente USA, Auburn MA) allowed for rapid imaging in Z every 1 μm over an 8-μm Z distance. NIS-Elements (Nikon, Melville, NY) controlled all equipment. All images were maximum intensity projections and processed using ImageJ/FIJI.

## Expression and purification of UBA1 enzymes

Recombinant UBA1b (aa 41–1058) protein was purified from Rosetta II (DE3) competent cells using previously established protocols (Beck et al, 2020). In brief, transformed *E.coli* were grown at 37 °C to an $OD_{600nm}$ of 1.5–2.0. Cultures were pre-chilled to 16 °C, followed by induction with 1 mM IPTG and incubation at 16 °C for ~16 h. Cells were collected and resuspended in lysis buffer (0.1 M Tris [pH 8.0], 0.5 M NaCl, 2 mM EDTA, 0.1% Triton-X100) supplemented with a protease inhibitor tablets (Roche). Bacteria were lysed using a LM10 microfluidizer at 15,000 psi and clarified by centrifugation at 4 °C, $50,000 \times g$ for 30 min. Clarified lysates were incubated with Ni-NTA agarose (Qiagen) at 4 °C for 1–2 h with rotation. Beads were washed with 10–15 column volumes of wash buffer (0.1 M Tris [pH 8.0], 0.5 M NaCl, 10 mM Imidazole/20 mM imidazole for abbreviated protocol), and subsequently eluted with 5–10 bead volumes of elution buffer (0.1 M Tris [pH 8.0], 0.3 M NaCl, 250 mM Imidazole). Ni-NTA eluates was concentrated using amicon ultra concentrators, MWCO 30 kDa (Millipore Sigma). For the abbreviated protocol, samples were desalted into storage buffer (50 mM Tris [pH 8.0], 0.2 M NaCl, 0.5 mM EDTA) using PD MiniTrap Sephadex G-25 columns (GE Healthcare). Otherwise, samples were further purified via gel filtration chromatography using a Superdex 200 Increase (GL 10/300) column. Samples were diluted into 0.1 M Tris [pH 8.0], concentrated, and further purified via anion exchange using a Mono Q (GL 5/50) column. All proteins from both methods were finally diluted into storage buffer (50 mM Tris [pH 8.0], 0.2 M NaCl, 25% glycerol), concentrated, aliquoted, and snap-frozen in liquid N2 to be stored at −80 °C.

## Adenylation activity and ATP kinetics

UBA1 adenylation and ATP Michaelis-Menten kinetics we determined using the EnzChek™ Pyrophosphate Assay Kit (ThermoFisher, E6645), following the manufacturer's protocol. In brief, activity of wild-type UBA1 and indicated mutants were measured by the release of $PP_i$, formed as a byproduct of ATP catalysis during ubiquitin-adenylate formation. Subsequently, $PP_i$ is enzymatically used to convert 7-methylthioguanosine (MesG) into a chromophoric guanine derivative visible at 360 nm. The rate of $PP_i$ breakdown directly correlates to UBA1 ATP catalysis rate. For adenylation activity, 50 μl reactions containing 0.1 mM Tris-HCl [pH 7.5], 20 mM $MgCl_2$, 150 mM $NH_2OH$, 0.03 U/ml purine nucleoside phosphorylase, 1 U/ml pyrophosphatase, 200 μM MesG, 2 mM ATP, 100 μM ubiquitin and varying quantities of recombinant UBA1b (3–400 nM) were prepared. Similarly, ATP Michaelis-Menten kinetics were performed in reactions containing 100 nM UBA1b and varying ATP concentrations (0–2000 μM). All buffers and reagents were phosphate and pyrophosphate free. Working stocks of hydroxylamine were prepared fresh every week. All reactions were performed in a Krystal™ 384-well glass bottom microplate (Southern Labware, 324021) and monitored every 30 s for 30 min, using a Synergy Neo2 multiplate reader (BioTek). The catalytic rate at each condition was determined by plotting the curves as $Abs_{360nm}$ vs. time and determining the rate of $PP_i$ production (slope) by a linear regression fit, followed by a conversion of $Abs_{360nm}$ into nM PPi using a standard curve. For adenylation activity, the catalytic rate vs. μg protein was plotted and adenylation activity was determined from the slope of the resulting linear regression. In the case of ATP titration, catalytic rate was plotted vs. [ATP] and fit to a Michalis-Menten kinetics model using GraphPad Prism (version 9).

## In vitro UBA1 ubiquitin thioester formation assays

To measure defects in UBA1 ubiquitin thioester formation, 0.5 μM UBA1 in reaction buffer (25 mM Tris-HCl [pH 8.0], 150 mM NaCl, 10 mM $MgCl_2$) was incubated with 10 μM ubiquitin and 10 mM ATP for 10 min at 37 °C or on ice (to slow down the reactions since the rate of ubiquitin thioester formation is very fast). Reactions were stopped via addition of 2X urea sample buffer (150 mM Tris [pH 6.5], 6 M urea, 6% SDS, 25% glycerol, <0.1% bromophenol blue) and UBA1-ubiquitin conjugates were separated by SDS-PAGE and detected via immunoblot analysis.

## In vitro thermolability assays

To observe the impact of temperature on ubiquitin thioester charging, samples of 0.5 μM UBA1 in reaction buffer (25 mM Tris-HCl [pH 8.0], 150 mM NaCl, 10 mM $MgCl_2$) with 10 μM monoubiquitin were pre-incubated for 30 min on ice or at 35, 37, 39, 41, or 43 °C. Next, 10 mM ATP was added and all reactions proceeded on ice. Reactions were stopped via addition of 2X urea sample buffer and UBA1-ubiquitin conjugates separated by SDS-PAGE and detected via immunoblot analysis.

## In vitro E2 ubiquitin transfer assays

To determine if mutations in UBA1 affect the transfer of ubiquitin to an E2, we devised a three-phase in vitro assay. (i) 0.25 μM UBA1 in reaction buffer (25 mM Tris-HCl [pH 8.0], 150 mM NaCl, 10 mM $MgCl_2$) was incubated with 10 μM ubiquitin and 10 mM ATP for 10 min at 30 °C. (ii) 100 mM EDTA was added for 10–30 min, and 1 μM indicated was added to initiate single transfer. (iii) 100 mM $MgCl_2$ was added to initiate multiple turnovers. After proceeding for 10–60 min, reactions were stopped via addition of 2× urea sample buffer and UBA1 and E2 charging were separated by SDS-PAGE and detected via immunoblot analysis. Alternatively, 10 μM FITC-ubiquitin was used in the assay and SDS-PAGE gels were analyzed by a fluorescence scanner (Chemidoc MP, Biorad).

## Structural modeling, molecular dynamics simulations, and $pK_a$ calculations

To generate a model structure of UBA1 A478S bound to Ub-AMP, the residue A444 in the crystal structure of *S. cerevisiae* UBA1 bound to Ub-AMP (PDB: 4NNJ) was mutated to serine and subsequently energy minimized by 500 iterations in Schrödinger (release 2022-2) with Maestro (Schrödinger Release 2017–4: Maestro. Schrödinger, LLC, New York, NY, 2017). Molecular dynamics simulations were performed in Schrödinger (release 2022-2) with Desmond. Structures of UBA1 from yeast (PDB: 6NYA and 4NNJ) and human (PDB: 6DC6) were prepared

identically. Briefly, sidechains and missing loops were modeled with Prime (Jacobson et al, 2004) and ionization states of sidechains were determined with PROPKA3 (Olsson et al, 2011). The system was solvated with the TIP3P water model applying a rhombic dodecahedron periodic boundary condition and neutralized with counterions to a final ionic strength of 150 mM. Equilibration was performed in the isothermal-isochoric (NVT) and isothermal-isobaric (NPT) ensemble at 300 K. Production simulations were performed in the NPT ensemble for 100 ns with a 2 fs timestep. The per residue root mean square fluctuations (r.m.s.f.) were calculated in Desmond across the entire trajectory and mapped onto the indicated crystal structures. From a 100 ns trajectory, distances between serine and the putative catalytic base were measured between the sidechain oxygens and plotted as a function of time with a moving window average of 100 ps. Structures were analyzed and figures generated by using PyMol (PyMOL Molecular Graphics System, http://www.pymol.org).

When calculating the $pK_a$ of serine, cysteine, and aspartic acid in the S621C variant, the yeast UBA1 structure (PDB: 4NNJ) was used. S621 was mutated to cysteine, simulated for 100 ns and the final frame was used for free energy calculations. FEP+ was used to calculate the free energy difference of protonation/deprotonation and the change in the $pK_a$ was calculated as a difference from a reference with the relationship below (Eq. 1) (Scarabelli et al, 2022). The reference model system is a three residue peptide identical to the sequence of the protein with the residue of interest in the center of this sequence. The reference $pK_a$ of this model system is assumed to be that of a typical sidechain at physiological pH. The free energy difference ($\Delta\Delta G_{calc}$) is that of the deprotonation occurring in the model system relative to this event occurring in the chemical environment of the protein. The values for R and T are defined as the ideal gas constant and temperature, respectively.

$$pK_{a,system} = pK_{a,reference} + \frac{\Delta\Delta G_{calc}}{2.303RT} \qquad (1)$$

### Reproducibility, quantification, and statistical analysis

Measures taken to verify reproducibility included to perform experiments at least three times (exact numbers of replicates and whether they are biological or technical is specified in each figure legend). No statistical analysis was performed to predetermine sample size, but three biological replicates is considered standard practice for most biochemical and cellular assays. No data was excluded from this study. Blinding was not used in our biochemical and cell biological study. Immunoblot quantifications were performed using Fiji. Details of replicates for each experiment are provided in the figure legends and the source data. Statistical analyses were performed with GraphPad Prism v.9. Data are presented as mean ± S.D. unless otherwise noted in figure legend. For comparisons between 3 or more groups, a one-way ANOVA with Tukey's multiple comparisons test was used.

## Data availability

All data supporting the findings are available in the main text and the appendix. Proteomics data are provided in Figures EV2, EV3

and were deposited into the Mass Spectrometry Interactive Virtual Environment (MassIVE) under the accession number MSV000092914. Requests for resources and reagents should be directed to and will be fulfilled by the lead contact, Achim Werner (Achim.werner@nih.com).

## Peer review information

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

## Acknowledgements

We thank Dr. Alan Schwartz from Washington University in St. Louis for the kind gift of the CHO cells. We thank the NIDCR Imaging Core (ZIC DE000744-04) and the Mass Spectrometry Facility (ZIA DE00075) for excellent technical assistance. We thank patients and their families for participation in our study. This research was supported by the Intramural Research Program of the National Institutes of Dental and Craniofacial Research (NIDCR, ZIA DE000749) and the National Cancer Institute (NCI, ZIA BC011490). DBB is supported by the Jeffrey Modell Foundation, the Relapsing Polychondritis Foundation, AAMDSIF, the Department of Defense (BMFRP- IDA HT9425-23-1-0507), and funding from the NIH (R00AR078205). This work was, in part, supported by computational resources from the NIH HPC Biowulf cluster (http://hpc.nih.gov).

## Author contributions

**Jason C Collins**: Conceptualization; Data curation; Formal analysis; Investigation; Writing—review and editing. **Samuel J Magaziner**: Formal

analysis; Investigation. **Maya English**: Data curation; Formal analysis; Investigation; Writing—review and editing. **Bakar Hassan**: Formal analysis; Investigation; Visualization; Writing—review and editing. **Xiang Chen**: Formal analysis; Investigation; Visualization. **Nicholas Balanda**: Formal analysis; Investigation. **Meghan Anderson**: Formal analysis; Investigation. **Athena Lam**: Formal analysis; Investigation. **Sebastian Fernandez-Pol**: Resources; Formal analysis; Investigation. **Bernice Kwong**: Formal analysis; Investigation. **Peter L Greenberg**: Formal analysis; Investigation. **Benjamin Terrier**: Formal analysis; Investigation. **Mary E Likhite**: Formal analysis; Investigation. **Olivier Kosmider**: Formal analysis; Investigation. **Yan Wang**: Resources; Formal analysis; Investigation; Writing—review and editing. **Nadine L Samara**: Formal analysis; Investigation. **Kylie J Walters**: Conceptualization; Formal analysis; Supervision; Funding acquisition; Investigation; Visualization; Writing—review and editing. **David B Beck**: Conceptualization; Resources; Data curation; Formal analysis; Supervision; Funding acquisition; Investigation; Writing—original draft; Writing —review and editing. **Achim Werner**: Conceptualization; Data curation; Formal analysis; Supervision; Funding acquisition; Investigation; Visualization; Writing —original draft; Writing—review and editing.

## Disclosure and competing interests statement
The authors declare no competing interests.

 

# Expanded View Figures

**Figure EV1.   Identification and clinical phenotypes of novel non-pMet41 VEXAS mutations.**

(A) Sanger sequencing confirming novel variants. (B) Digital droplet PCR (ddPCR) confirmation for novel variants in P3, P4, P5. $n = 3$ technical replicates, error bars = s.d. (C) Conservation of protein sequence for UBA1. (D) Cytoplasmic vacuoles were seen in a subset of the proerythroblasts (Panel i, ii, iii) and promyelocytes (Panel iii, iv, v, vi) from P6. Scale bar = 20 μm. (E) Sweets syndrome in P6.

▶

     *The EMBO Journal* Volume 43 | Issue 10 | May 2024 | 1919 – 1946     **1941**

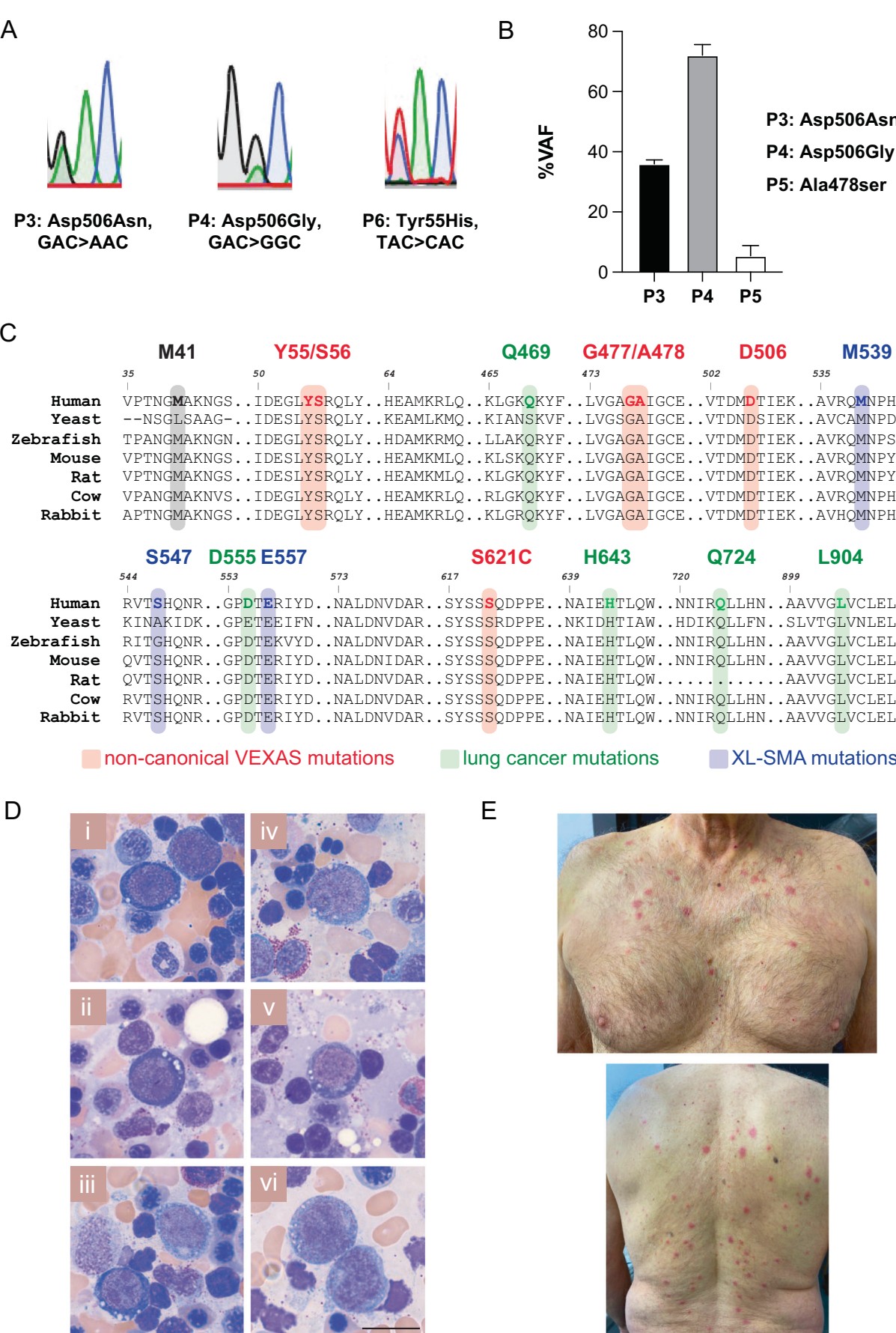

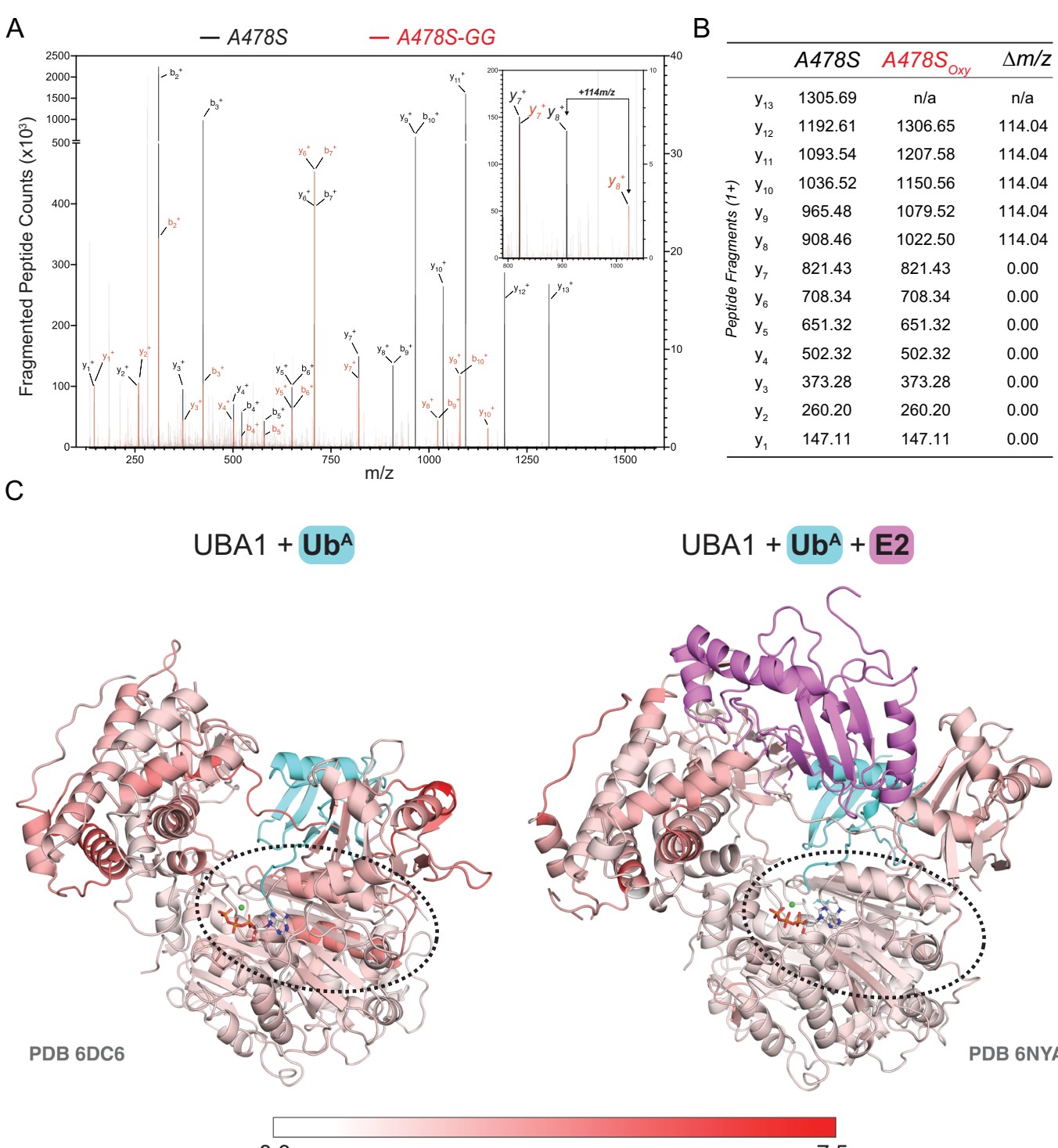

**Figure EV2.  UBA1 p.A478S forms an aberrant oxyester at the mutation site.**

(**A**) Annotated MS/MS spectrum of A478S peptide with (red) and without (black) diGly remnant. Insert highlights the difference of 114 Da, indicative of the diGly remnant. (**B**) Table summarizing the masses of fragments of the unmodified and ubiquitylated A478S peptide, pinpointing the diGly remnant ($\Delta m/z$ 114.04 Da) on S478. (**C**) Side-by-side ribbon diagrams of UBA1 bound to ubiquitin (cyan) and ATP (indicated as a stick diagram with carbon, oxygen, nitrogen, and phosphorus in gray, red, blue, and orange) following 100 ns of molecular dynamics in the absence (left, PDB: 6DC6) or presence (right, PDB: 6NYA) of E2 enzyme (Ubc3, purple). The per residue r.m.s.f. (root mean square fluctuation) value is indicated as a red gradient (scale bar). A black dashed circle highlights the region where ubiquitin, ATP, and Mg$^{2+}$ (green sphere) bind.

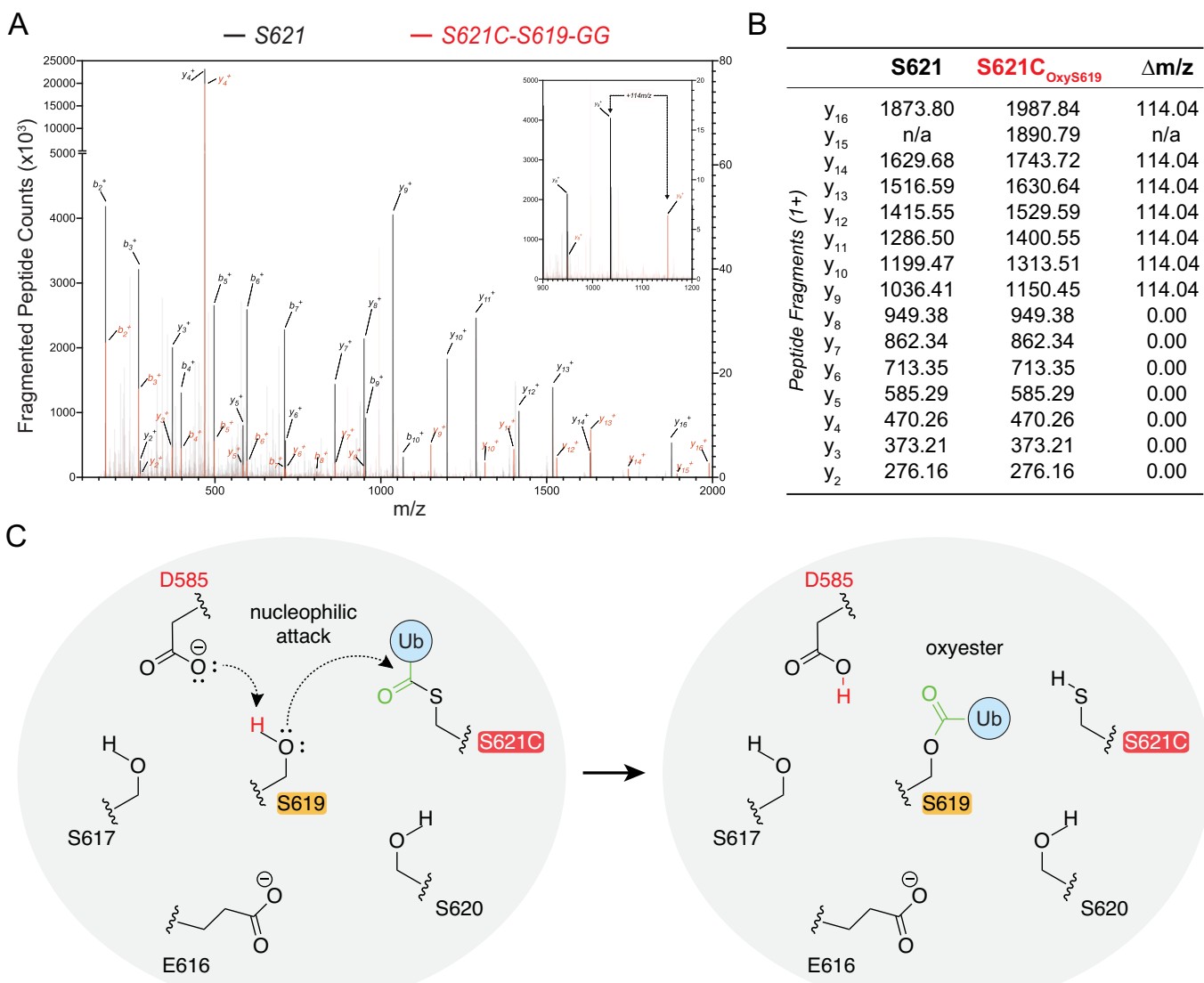

| | S621 | S621C$_{OxyS619}$ | Δm/z |
|---|---|---|---|
| $y_{16}$ | 1873.80 | 1987.84 | 114.04 |
| $y_{15}$ | n/a | 1890.79 | n/a |
| $y_{14}$ | 1629.68 | 1743.72 | 114.04 |
| $y_{13}$ | 1516.59 | 1630.64 | 114.04 |
| $y_{12}$ | 1415.55 | 1529.59 | 114.04 |
| $y_{11}$ | 1286.50 | 1400.55 | 114.04 |
| $y_{10}$ | 1199.47 | 1313.51 | 114.04 |
| $y_{9}$ | 1036.41 | 1150.45 | 114.04 |
| $y_{8}$ | 949.38 | 949.38 | 0.00 |
| $y_{7}$ | 862.34 | 862.34 | 0.00 |
| $y_{6}$ | 713.35 | 713.35 | 0.00 |
| $y_{5}$ | 585.29 | 585.29 | 0.00 |
| $y_{4}$ | 470.26 | 470.26 | 0.00 |
| $y_{3}$ | 373.21 | 373.21 | 0.00 |
| $y_{2}$ | 276.16 | 276.16 | 0.00 |

**Figure EV3.   UBA1 p.S621C forms an aberrant oxyester at S619 via a thioester intermediate at S621C.**

(A) Annotated MS/MS spectrum of S621C peptide with (red) and without (black) diGly remnant. Insert highlights the difference of 114 Da, indicative of the diGly remnant. (B) Table summarizing the masses of fragments of the unmodified and ubiquitylated S621C peptide, pinpointing the diGly remnant (Δm/z 114.04 Da) on S619. (C) Proposed catalytic mechanism for oxyester formation through deprotonation of S619 via the putative catalytic base D585 forming an oxyanion that promotes a nucleophilic attack of the thioester at S621C.

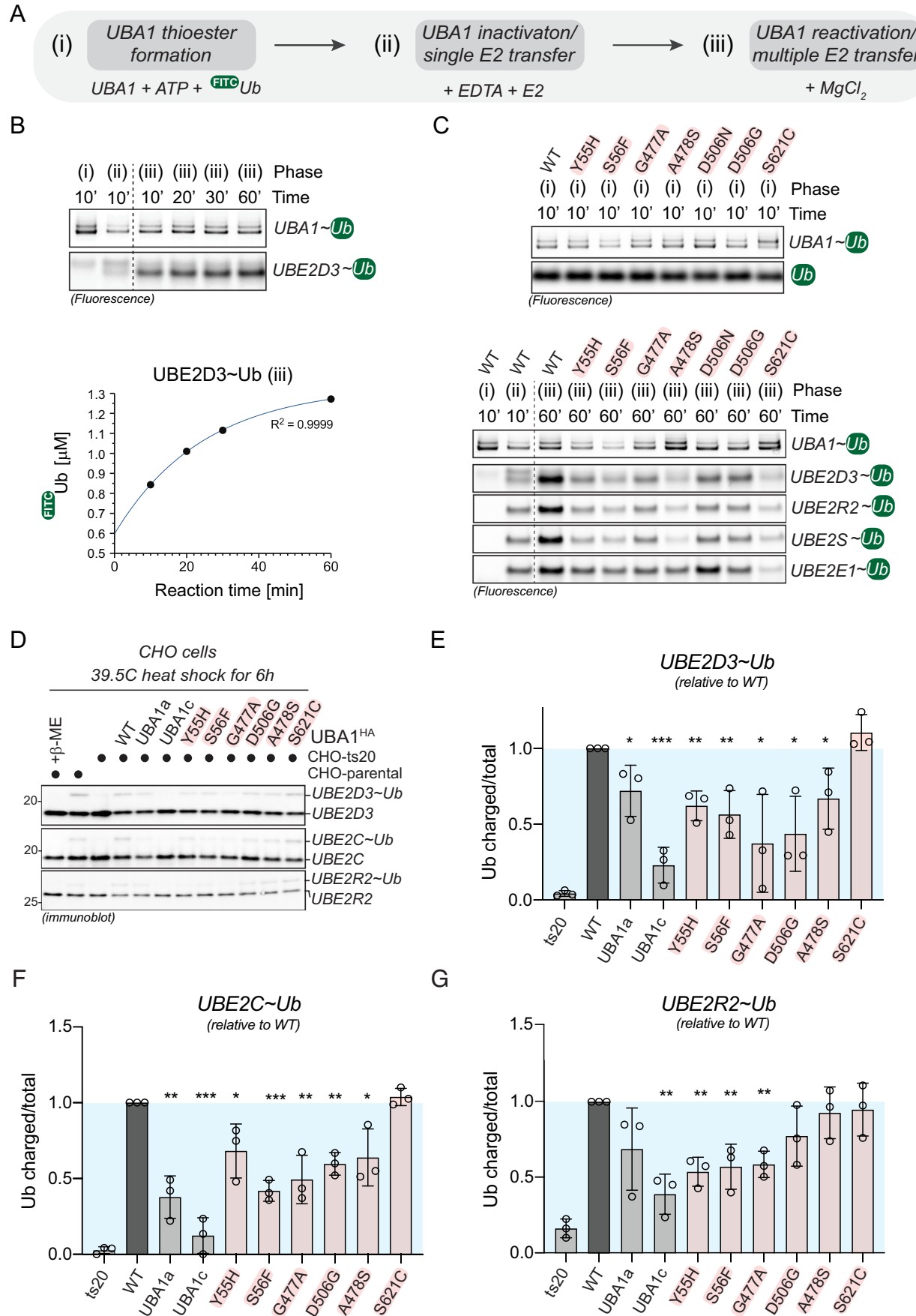

**Figure EV4.   Non-canonical VEXAS mutations are deficient in transferring ubiquitin to diverse E2 enzymes in vitro and exhibit defects in cells.**

(A) Schematic overview of the sequential, three-phase in vitro assay used to measure UBA1 transthiolation. (i) Complete charging of UBA1 by incubation of 250 nM UBA1b with 10 µM FITC-ubiquitin and 5 mM ATP (ii) Quenching of UBA1 charging and single transfer to E2 enzyme by addition of 100 mM EDTA and 1 µM E2 enzyme (iii) Reactivation of UBA1 charging and multi-transfer to E2 enzyme by addition of 100 mM MgCl₂ (B) UBA1 WT was subjected to the experiment described in panel (A) using UBE2D3. Reactions were subjected to SDS page and analyzed by fluorescence imaging. *Upper panel*: Fluorescence scan showing UBA1 and UBE2D3 ubiquitin thioester levels after each reaction phase. *Lower panel*: UBE2D3 ubiquitin thioester levels in reaction phase (iii) were quantified and plotted against the reaction time, revealing that UBE2D3 is maximally charged after 60 min. (C) Non-canonical VEXAS mutations (red) are deficient in E2 transthiolation in vitro. Indicated UBA1 proteins were subjected to the experiment described in panel (A) using either UBE2D3, UBE2R2, or UBE2S. Reactions were subjected to SDS page and analyzed by fluorescence imaging. *Upper panel*: Fluorescence scan showing UBA1 charging after reaction phase (i) as control. *Lower panel*: Fluorescence scan showing UBA1 re-charging and E2 transfer after reaction phase (iii). Quantifications of 3 biological replicates are shown in Fig. 5F. (D) Non-canonical VEXAS mutations (red) are impaired in supporting E2 ubiquitin thioester levels in cells. CHO ts20 cells were reconstituted with indicated UBA1 variants and incubated at the permissive temperature for 6 h, followed by immunoblotting using antibodies against indicated E2 enzymes. (E–G) Quantification of ubiquitin charging levels of indicated E2s (charged/total) shown in panel (D). $n = 3$ biological replicates, error bars = s.d., *$p < 0.05$, **$p < 0.01$, ***$p < 0.001$, one-way ANOVA.

     