## [Peer Review File · The EMBO Journal]

Shared and Distinct Mechanisms of UBA1 Inactivation Across Different Diseases

Jason Collins, Samuel Magaziner, Maya English, Bakar Hassan, Xiang Chen, Nicholas Balanda, Meghan Anderson, Athena Lam, Sebastian Fernandez-Pol, Bernice Kwong, Peter Greenberg, Benjamin Terrier, Mary Likhite, Olivier Kosmider, Yan Wang, Nadine Samara, Kylie Walters, David Beck, and Achim Werner

Corresponding author(s): Achim Werner (achim.werner@nih.gov)

Review Timeline:

Submission Date:	22nd Sep 23
Editorial Decision:	7th Nov 23
Revision Received:	13th Dec 23
Editorial Decision:	11th Jan 24
Revision Received:	25th Jan 24
Accepted:	29th Jan 24

Editor: Hartmut Vodermaier

Transaction Report:

Dr. Achim Werner
NIDCR/NIH
Stem Cell Biochemistry Unit
30 CONVENT DR MSC 4340
Bethesda, MD 20892-4340

7th Nov 2023

Re: EMBOJ-2023-115688
Shared and Distinct Mechanisms of UBA1 Inactivation Across Different Diseases

Dear Achim,

Thank you again for submitting your study of disease-associated UBA1 variants and their effects for our editorial consideration. I apologize for the delay in its evaluation - we sent it to three expert reviewers, but have so far only received the reports of two of them. Both of them agree on the overall interest as well as the quality of the work, and only raise a number of specific and mostly presentational concerns. In order to prevent unnecessary further loss of time, I have therefore decided to move forward at this stage, and to invite you to revise the manuscript based on these two attached reports.

Detailed information on preparing, formatting and uploading a revised manuscript can be found below and in our Guide to Authors; adhering to these guidelines as closely as possible shall greatly facilitate the editorial checking and processing at the time of resubmission - in particular regarding the completion of our author checklist, the inclusion of editable text files and individual figures, and the conversion of "supplemental" material into Expanded View and/or Appendix content. Please be reminded that it is our policy to allow only a single round of (major) revision, making it important to carefully respond to all points raised at this point. As per our 'scooping protection' policy, related or competing work appearing during the course of the revision period will not affect our final decision on your study. Should you have additional questions linked to this decision, the referee reports, or the revision guidelines, please do not hesitate to contact me.

Thank you again for the opportunity to consider this study for The EMBO Journal! I look forward to receiving your revision in due time.

With kind regards,

Hartmut

9) Digital image enhancement is acceptable practice, as long as it accurately represents the original data and conforms to community standards. If a figure has been subjected to significant electronic manipulation, this must be clearly noted in the figure legend and/or the 'Materials and Methods' section. The editors reserve the right to request original versions of figures and the original images that were used to assemble the figure. Finally, we generally encourage uploading of numerical as well as gel/blot image source data; for details see: embopress.org/page/journal/14602075/authorguide#sourcedata

At EMBO Press, we ask authors to provide source data for the main manuscript figures. Our source data coordinator will contact you to discuss which figure panels we would need source data for and will also provide you with helpful tips on how to upload and organize the files.

In the interest of ensuring the conceptual advance provided by the work, we recommend submitting a revision within 3 months (5th Feb 2024). Please discuss the revision progress ahead of this time with the editor if you require more time to complete the revisions. Use the link below to submit your revision:

Link Not Available

Referee #2:

Collins et al present an exceptionally thorough study of effects of disease-associated UBA1 mutations on cellular and biochemical functions.

Figure 1 builds on recent work from Beck and Werner on VEXAS syndrome, which has brought to the forefront the concept that the essential gene at the foundation of most cellular ubiquitylation is mutated in disease. The authors identify additional UBA1 mutations in patients with severe inflammation. These impact on isoforms produced through ectopic expression in HEK 293T cells, and in the absence of UBA1 at the nonpermissive temperature in ts CHO cells. There is an impact on total polyubiquitinated proteins in this setting.

The authors assay the different functions of E1 in vitro in the context of isoform b expressed in E coli. There are effects of the mutation on the kinetics of the adenylation reaction, on formation of the ubiquitin thioester intermediate with E1, and on formation of E2 ubiquitin thioester complexes in vitro and in cells. An unexpected finding is that some mutations cause ubiquitin linkage to serines detected by band shifting in the presence of hydroxylamine and identified by mass spec. A478S forms an oxyester at the mutation site. S621C promotes formation of oxyester at nearby S619. Finally, the authors also perform their assays on UBA1 mutations that segregate with SMA and are associated with lung cancer in never smokers.

Overall, the study is interesting, The experiments are performed to a high standard. The manuscript is a pleasure to read and the figures are both clear and beautiful. I have a few minor suggestions for the text and figures, and recommend publication in EMBO J.

Minor comments:

1. It would be good to include the isoform expressed in *E. coli* and used for the biochemical assays in the main text.
2. The authors may wish to make it more clear that the rate of ubiquitin thioester formation to E1 is extremely fast and it is difficult to assay for defects, which is why they needed to slow down the reaction. One question is why in Fig. 2g for S621C it looks like a substantial loss in the gel at 37C but not reflected in the bar graph. The authors might want to select more representative gels.
3. Although the author's interpretation that oxyester formation proceeds through the thioester intermediate makes sense, it is nearly impossible to see buildup and loss of the intermediate during temporal reaction so this wasn't assayed. The authors may wish to consider how to phrase this.
4. The authors should represent that the pKa is from simulations/calculations/predictions and not actually measured.
5. Is it absolutely clear that E1 mutations cause SMA? I looked at the reference and it seems that this mutation segregates with disease but am not certain causation is known. The authors should consider rewording.
6. Since isoform distribution could play a major role in function/dysfunction of the disease mutants, the authors should add this to Figure 8 if possible.

Referee #3:

In this manuscript, Collins and colleagues examined the mechanistic basis of disease-causing mutations in UBA1. UBA1 is a key mediator of ubiquitin signaling and disruption in UBA1 caused by somatic mutations, even in a small subset of cells, can result in very dramatic disease phenotypes. VEXAS (vacuoles, E1, X-linked, autoinflammatory, somatic) syndrome, previously described by some of the same authors, is a fascinating disease with a broad disease spectrum in itself. Variants in UBA1 has also been linked to lung cancer and spinal muscular atrophy. Aside from the predominant mutation at Met41 associated with VEXAS, little is known about how UBA1 variants affect gene function and contribute to pathology.

This manuscript provides a very comprehensive view of the different UBA1 variants associated with VEXAS, lung cancer and SMA. They found that only the Met41 variants affected transcription of a cytoplasmic isoform while other VEXAS variants and lung cancer variants affected transthioesterification. In contrast, the few SMA associated mutations seemed to be thermolabile. This work improves the knowledge of the structural to function relationship of UBA1, and provides insight to the association with various diseases. The experiments are rigorous, and the manuscript is generally well written, although perhaps a bit dense for readers that are less familiar with ubiquitination. The schematics with the figures are helpful to follow along the series of complex experiments. Overall, the work is important and advances our current understanding of the field.

This reviewer has a few minor suggestions for the authors to consider.

1. The *in vitro* and cellular system are supportive of the author's claims. However, I wonder if tissue specificity should be taken into consideration when addressing genotype phenotype correlation. VEXAS is a disorder of myeloid cells, which is quite different from lung cancer and SMA. There is the possibility that the VEXAS mutations won't have as much impact in the affected cells in lung cancer and SMA. Addressing this issue experimentally is complex and would be beyond the scope of the paper, but this should be discussed as a limitation.
2. Beyond an association, the connection between mutation mechanisms and three disease phenotypes is unresolved. This work describes different ways to result in UBA1 loss of function. At least in the context of VEXAS, inflammation is a hallmark of the disease. Do the authors have insight on whether mutations that are thermolabile (therefore less associated with VEXAS) also result in inflammation?
3. This paper discusses many UBA1 variants related to various disease phenotypes. It would be helpful to have a supplemental table to reference with the various figures - mutation, frequency, associated phenotype, some assessment of pathogenicity by CADD or equivalent prediction algorithm.
4. In many figure panels, some variants are highlighted in various colors. In many instances the reason for the highlight is not clear from the text or legend. Please clarify.

Point-by-point response to reviewer's comments:**Referee #2:**

Collins et al present an exceptionally thorough study of effects of disease-associated UBA1 mutations on cellular and biochemical functions.

Figure 1 builds on recent work from Beck and Werner on VEXAS syndrome, which has brought to the forefront the concept that the essential gene at the foundation of most cellular ubiquitylation is mutated in disease. The authors identify additional UBA1 mutations in patients with severe inflammation. These impact on isoforms produced through ectopic expression in HEK 293T cells, and in the absence of UBA1 at the nonpermissive temperature in ts CHO cells. There is an impact on total polyubiquitinated proteins in this setting.

The authors assay the different functions of E1 in vitro in the context of isoform b expressed in E coli. There are effects of the mutation on the kinetics of the adenylation reaction, on formation of the ubiquitin thioester intermediate with E1, and on formation of E2 ubiquitin thioester complexes in vitro and in cells. An unexpected finding is that some mutations cause ubiquitin linkage to serines detected by band shifting in the presence of hydroxylamine and identified by mass spec. A478S forms an oxyester at the mutation site. S621C promotes formation of oxyester at nearby S619. Finally, the authors also perform their assays on UBA1 mutations that segregate with SMA and are associated with lung cancer in never smokers.

Overall, the study is interesting, The experiments are performed to a high standard. The manuscript is a pleasure to read and the figures are both clear and beautiful. I have a few minor suggestions for the text and figures, and recommend publication in EMBO J.

We thank the reviewer for their positive evaluation of our study as well as their suggestions and comments, which we have addressed as described below.

Minor comments:

1. It would be good to include the isoform expressed in E. coli and used for the biochemical assays in the main text.

We agree with the reviewer and have now updated the main text and material and methods section to delineate that our study used UBA1b specifically, which is UBA1 aa 41-1058.

2. The authors may wish to make it more clear that the rate of ubiquitin thioester formation to E1 is extremely fast and it is difficult to assay for defects, which is why they needed to slow down the reaction. One question is why in Fig. 2g for S621C it looks like a substantial loss in the gel at 37C but not reflected in the bar graph. The authors might want to select more representative gels.

We agree with the reviewer and have included a statement in the material and method section explaining that reactions were slowed down on ice because at higher temperature the thioester formation occurs too rapidly to observe defects. We also now show a more representative immunoblot for the S621C mutant in figure 2G (n=5).

3. Although the author's interpretation that oxyester formation proceeds through the thioester intermediate makes sense, it is nearly impossible to see buildup and loss of the intermediate during temporal reaction so this wasn't assayed. The authors may wish to consider how to phrase this.

We agree with the reviewer that due to the transient nature of the thioester intermediates, it is extremely difficult to visualize their build up and loss during *in vitro* reactions. We have acknowledged this fact in the text of the revised manuscript. However, for both, UBA1 A478S and S621C, we believe that through combining mutational analyses, mass spectrometry, and molecular dynamics simulations, we have provided sufficient evidence for the proposed underlying mechanisms of oxyester formation (**Figure 3B-H; Figure 4B-H**).

4. The authors should represent that the pKa is from simulations/calculations/predictions and not actually measured.

We agree with the reviewer, and mentioned in the results part the pKa values were calculated. To further delineate this point and highlight that these are predicted pKAs, we have now edited the table in Figure 4G in the revised version of our manuscript.

5. Is it absolutely clear that E1 mutations cause SMA? I looked at the reference and it seems that this mutation segregates with disease but am not certain causation is known. The authors should consider rewording.

UBA1 mutations have been reported in X linked SMA in multiple publication (*Ramser et al., AJHG, 2009, PMID: 18179898*). Additionally, morpholino studies in zebrafish have demonstrated that loss of Uba1 leads to neurodevelopmental defects (*Powis et al, JCI Insights, 2016, PMID: 27699224*). Although these studies are limited by either a small patient population, or lack of specific mutation testing, we think that there is sufficient evidence to support the role of UBA1 in SMA. To address that lack of definitive proof with specific knock-in animal models, we have changed the text to say SMA-associated mutations throughout the manuscript.

6. Since isoform distribution could play a major role in function/dysfunction of the disease mutants, the authors should add this to Figure 8 if possible.

We thank the reviewer for this comment. In Appendix Figure S1 of the revised version of our manuscript (Supplementary Figure 2 in the old version of the manuscript), we have added immunofluorescence for SMA- and lung cancer-associated UBA1 mutations and do not identify any obvious alterations in subcellular localization. We have also included

this data in our summary heat map in Figure 8.

Referee #3:

In this manuscript, Collins and colleagues examined the mechanistic basis of disease-causing mutations in UBA1. UBA1 is a key mediator of ubiquitin signaling and disruption in UBA1 caused by somatic mutations, even in a small subset of cells, can result in very dramatic disease phenotypes. VEXAS (vacuoles, E1, X-linked, autoinflammatory, somatic) syndrome, previously described by some of the same authors, is a fascinating disease with a broad disease spectrum in itself. Variants in UBA1 has also been linked to lung cancer and spinal muscular atrophy. Aside from the predominant mutation at Met41 associated with VEXAS, little is known about how UBA1 variants affect gene function and contribute to pathology.

This manuscript provides a very comprehensive view of the different UBA1 variants associated with VEXAS, lung cancer and SMA. They found that only the Met41 variants affected transcription of a cytoplasmic isoform while other VEXAS variants and lung cancer variants affected transthioesterification. In contrast, the few SMA associated mutations seemed to be thermolabile. This work improves the knowledge of the structural to function relationship of UBA1, and provides insight to the association with various diseases. The experiments are rigorous, and the manuscript is generally well written, although perhaps a bit dense for readers that are less familiar with ubiquitination. The schematics with the figures are helpful to follow along the series of complex experiments. Overall, the work is important and advances our current understanding of the field.

This reviewer has a few minor suggestions for the authors to consider.

We thank the reviewer for their positive evaluation of our study as well as their suggestions and comments, which we have addressed as described below.

Minor comments:

1. The in vitro and cellular system are supportive of the author's claims. However, I wonder if tissue specificity should be taken into consideration when addressing genotype phenotype correlation. VEXAS is a disorder of myeloid cells, which is quite different from lung cancer and SMA. There is the possibility that the VEXAS mutations won't have as much impact in the affected cells in lung cancer and SMA. Addressing this issue experimentally is complex and would be beyond the scope of the paper, but this should be discussed as a limitation.

We completely agree with the reviewer that tissue specificity may drive some of the disease and mutation specific manifestation. We are eager to address these questions in future experiments. We clearly articulate these limitations and emerging questions in the last paragraph of the discussion, which we have now expanded.

2. Beyond an association, the connection between mutation mechanisms and three disease phenotypes is unresolved. This work describes different ways to result in UBA1 loss of function. At least in the context of VEXAS, inflammation is a hallmark of the

disease. Do the authors have insight on whether mutations that are thermolabile (therefore less associated with VEXAS) also result in inflammation?

We thank the reviewer for raising this intriguing question. To date, thermolabile mutations associated with lung cancer in never smokers and SMA have not been observed in VEXAS patients. Thus, it remains unclear whether these mutations would result in inflammation if present in the myeloid lineage. We are in the process of developing relevant myeloid cell models and are excited to address this question experimentally in the future, as results from these studies could inform on tissue-specific functions of UBA1. In the revised version of our manuscript, we have added a sentence to discuss this point in the last paragraph of the discussion.

3. This paper discusses many UBA1 variants related to various disease phenotypes. It would be helpful to have a supplemental table to reference with the various figures - mutation, frequency, associated phenotype, some assessment of pathogenicity by CADD or equivalent prediction algorithm.

We agree with the reviewer and have provided such a table listing all UBA1 mutations studied in this manuscript in the new supplemental table 2. We reference this table in the manuscript when we introduce the non-canonical VEXAS mutations and the SMA and lung cancer-associated mutations.

4. In many figure panels, some variants are highlighted in various colors. In many instances the reason for the highlight is not clear from the text or legend. Please clarify.

We thank the reviewer for pointing out these oversights. Throughout the manuscript, we have labeled non-canonical VEXAS mutations in red, LCINS-associated mutations in green, and SMA-associated mutations in blue. We have also highlighted other residues of UBA1 in different colors (e.g. the catalytic cysteine residue C632 in yellow and the oxyester formation site S619 in UBA1 S621C in orange). In the revised version of our manuscript, we have now updated all figure legends to define this color-coding scheme.

Dr. Achim Werner
NIDCR/NIH
Stem Cell Biochemistry Unit
30 CONVENT DR MSC 4340
Bethesda, MD 20892-4340

11th Jan 2024

Re: EMBOJ-2023-115688R
Shared and Distinct Mechanisms of UBA1 Inactivation Across Different Diseases

Dear Achim,

Thank you for submitting your revised manuscript to The EMBO Journal. I have now carefully checked your responses to the original comments, and referee 3 (see feedback below) has also looked at the manuscript once more. Given that all points have been satisfactorily addressed, we shall be happy to accept the study for publication, as soon as a few remaining editorial issues have been taken care of:

- Please upload separate text and figure files (1 file per each main and 1 per each EV figure). Figure legends should be at the end of the main text file.
- On the abstract page of the manuscript, please reduce the number of keyword terms to 4-5, preferentially choosing general terms.
- Please double-check all citations in the reference list, as some of them appear to be incomplete (lacking page/locator numbers). Also, please change the section header to "References"
- Please rename the Conflict of Interest section into "Disclosure and Competing Interests Statement", in accordance with our updated Guide to Authors (<https://www.embopress.org/competing-interests/>)
- As we are switching from a free-text author contribution statement towards a more formal statement based on Contributor Role Taxonomy (CRediT) terms, please remove the present Author Contribution section and instead specify each author's contribution(s) directly in the Author Information page of our submission system during upload of the final manuscript. See <https://casrai.org/credit/> for more information.
- Please remove reviewer access information from the Data Availability section at this point, and ensure that data become publicly accessible upon acceptance.
- Figure 4G has not been called out in the manuscript, please check (most likely, it has been mistakenly referenced as 5G?).
- For the 2 "Expanded View" tables, please include their legends in a separate tab of each of the XLSX spreadsheets. Also, given that "Table EV2" has multiple tabs, it should be renamed as "Dataset EV1", making sure that also its callouts in the text (and the naming inside the file) are updated accordingly.
- Please double-check to make sure to all relevant funding information in the manuscript is congruent with the info entered into our submission system. (Missing in the system currently: the Jeffrey Modell Foundation, the Relapsing Polychondritis Foundation, AAMDSIF, the Department of Defense (BMFRP- IDA HT9425-23-1-0507).
- Please provide suggestions for a short 'blurb' text prefacing and summing up the conceptual aspect of the study in two sentences (max. 250 characters), followed by 3-5 one-sentence 'bullet points' with brief factual statements of key results of the paper; they will form the basis of an editor-written 'Synopsis' accompanying the online version of the article. Please also upload a synopsis image, which can be used as a "visual title" for the synopsis section of your paper. The image should be in PNG or JPG format, and please make sure that it remains in the modest dimensions of (exactly) 550 pixels wide and 300-600 pixels high.
- Finally, during routine pre-acceptance checks, our data editors have raised the following queries regarding figures, data, and legends, which I would ask you to address (ideally using track changes again):
 - * Please note that in figure 1f; there is a mismatch between the annotated p values in the figure legend and the annotated p values in the figure file that should be corrected.
 - * Please note that information related to n is missing in the legends of figure 4f; EV 1b; EV 4e-g.
 - * Please note that the error bars are not defined in the legends of figure 4f; EV 1b; Ev 4e-g.

* Please note that the measure of center for the error bars needs to be defined in the legends of figures 1f; 2d-e, h; 5c-e; 6d-f; 7f.

* Please note that scale bar and its definition are missing for figure EV 1d.

I am therefore returning the manuscript to you for a final round of minor revision, to allow you to make these modifications and upload the revised files. Once we will have received them, we should be ready to swiftly proceed with formal acceptance and production of the manuscript.

With best regards,

Hartmut

9) Digital image enhancement is acceptable practice, as long as it accurately represents the original data and conforms to community standards. If a figure has been subjected to significant electronic manipulation, this must be clearly noted in the figure legend and/or the 'Materials and Methods' section. The editors reserve the right to request original versions of figures and the original images that were used to assemble the figure. Finally, we generally encourage uploading of numerical as well as gel/blot image source data; for details see: embopress.org/page/journal/14602075/authorguide#sourcedata

At EMBO Press, we ask authors to provide source data for the main manuscript figures. Our source data coordinator will contact you to discuss which figure panels we would need source data for and will also provide you with helpful tips on how to upload and organize the files.

Further information is available in our Guide For Authors:

In the interest of ensuring the conceptual advance provided by the work, we recommend submitting a revision within 3 months (10th Apr 2024). Please discuss the revision progress ahead of this time with the editor if you require more time to complete the revisions. Use the link below to submit your revision:

Link Not Available

Referee #3:

The authors have addressed my critiques. I congratulate them on this impressive study.

Dr. Achim Werner
NIDCR/NIH
Stem Cell Biochemistry Unit
30 CONVENT DR MSC 4340
Bethesda, MD 20892-4340

29th Jan 2024

Re: EMBOJ-2023-115688R1
Shared and Distinct Mechanisms of UBA1 Inactivation Across Different Diseases

Dear Achim,

Thank you for submitting your final revised manuscript for our consideration. I am pleased to inform you that we have now accepted it for publication in The EMBO Journal.

With kind regards,

Hartmut
